# L2P-MIP: Learning to Presolve for Mixed Integer Programming

**Chang Liu**[1], **Zhichen Dong**[1], **Haobo Ma**[1], **Weilin Luo**[2], **Xijun Li**[2], **Bowen Pang**[2], **Jia Zeng**[2], **Junchi Yan**[1]*

[1]Department of Computer Science and Engineering, Shanghai Jiao Tong University
[2]Huawei Noah's Ark Lab
`{only-changer,niconi19,witcher,yanjunchi}@sjtu.edu.cn`
`{luoweilin3,pangbowen2,xijun.li,zeng.jia}@huawei.com`
PyTorch Code: https://github.com/Thinklab-SJTU/L2P-MIP

## Abstract

Modern solvers for solving mixed integer programming (MIP) often rely on the branch-and-bound (B&B) algorithm which could be of high time complexity, and presolving techniques are well designed to simplify the instance as pre-processing before B&B. However, such presolvers in existing literature or open-source solvers are mostly set by default agnostic to specific input instances, and few studies have been reported on tailoring presolving settings. In this paper, we aim to dive into this open question and show that the MIP solver can be indeed largely improved when switching the default instance-agnostic presolving into instance-specific presolving. Specifically, we propose a combination of supervised learning and classic heuristics to achieve efficient presolving adjusting, avoiding tedious reinforcement learning. Notably, our approach is orthogonal from many recent efforts in incorporating learning modules into the B&B framework after the presolving stage, and to our best knowledge, this is the first work for introducing learning to presolve in MIP solvers. Experiments on multiple real-world datasets show that well-trained neural networks can infer proper presolving for arbitrary incoming MIP instances in less than 0.5s, which is neglectable compared with the solving time often hours or days.

## 1 Introduction and Related Work

Mixed integer programming (MIP) is a general optimization formulation of various real-world optimization applications, such as scheduling and production planning. In its commonly studied linear form, MIP minimizes a linear objective function over a set of integer points that satisfy a finite family of linear constraints. Due to its NP-hard nature, in modern MIP solvers (SCIP (Gamrath et al., 2020), GUROBI (Gurobi, 2021), CPLEX (IBM, 2021)), the branch-and-bound (B&B) algorithm is widely employed. B&B traverses the candidate solutions systematically, in which the set of candidate solutions is considered to form a search tree with the full set at the root. However, B&B can suffer from severe scalability issues in branching selections, especially for real-world applications.

Efforts have been made to reduce the time cost of B&B by including an extra step: given an MIP instance, the solver first pre-processes and simplifies the instance before passing it to B&B. This step is usually named **Presolve**, and various presolvers have been designed to reduce the size of the input instance. Via presolving, the original MIP instance is simplified by removing irrelevant information e.g. redundant constraints and variables. After presolving, B&B only needs to solve the smaller simplified instance. Though the presolving step itself does cost extra time, it leads to a great time saving for the B&B algorithm, and in total improves the performance of the MIP solver significantly (Achterberg et al., 2019). It has been shown in early studies (Bixby et al., 2004; Achterberg & Wunderling, 2013b) that, after appropriately presolving, 1.3x speed up can be acquired and more than 15% unsolvable instances become solvable within the time limit. Due to the page limit, we place the description of the commonly used presolvers in the Appendix (A.1).

*Correspondence author. The work was in part supported by National Key Research and Development Program of China (2020AAA0107600), Huawei Technologies, NSFC (62222607), and SJTU Trans-med Awards Research (STAR) 20210106.

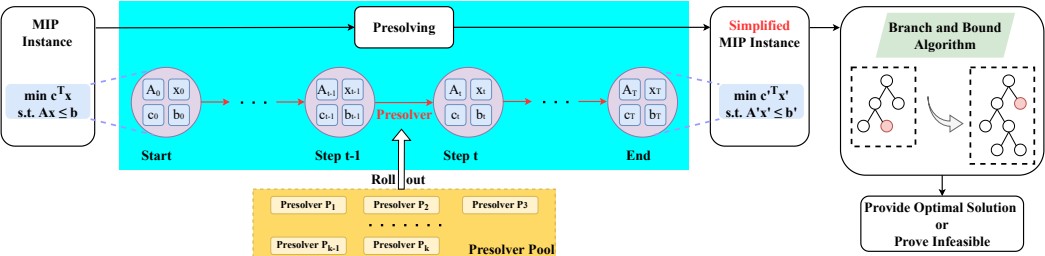

Figure 1: Presolving in MIP solvers. For each incoming MIP instance, the solver first presolves it to simplify the problem, which includes multiple rounds and the utilization of multiple presolvers. Then, the simplified instance is passed to the B&B algorithm and solved.

In existing MIP solvers, presolving is routinely adopted by the default setting, being agnostic to input instances. As the default setting may not always be suitable, several works (Hutter et al., 2009; 2011; Lindauer et al., 2022) propose to find one single robust configuration (setting) for all problem instances. However, they still can not tailor presolving for each unseen instance. In this paper, we argue that tailoring suitable presolving for each individual instance could reach better performances. Researchers of the latest work (Galabova, 2023) conclude that "*an analysis of presolve would be incomplete without an investigation of this effect for particular instances*", which necessitates instance-specific tailoring presolving. Moreover, the value of instance-specific tailoring in presolving has been empirically shown in (Frank et al., 2010).

To this end, we try to design an efficient method that can tailor presolving for MIP instances, which is able to integrate into existing MIP solvers. In general, customized presolving includes how to pick the next presolver (determine the order), limit the max used rounds of each presolver, and set the time used for each presolver. Especially, some of the operations can be related to each other, for example, the order of the presolvers can influence their utilization rate and efficiency. Hence, finding the best presolving is a challenging task and remains under-studied in literature.

To achieve instance-adaptive presolving, one provable way is using heuristic algorithms to search for the best presolving. However, heuristic searching can be too time-consuming to serve as a pre-processor. To improve efficiency, neural networks can be used to fit the behavior of heuristic algorithms, since neural networks can infer the suitable presolving in a short time. More specifically, by taking a closer look at the presolving parameters, we argue that the priority is more influential and can affect the other parameters since priority determines the execution order of presolvers. As shown in many previous works (Elble, 2010; Lodi & Tramontani, 2013; Galabova, 2023), the performance of presolving is very sensitive to the order of the presolvers.

In this paper, we propose a hybrid algorithmic neural framework for improving presolving, namely **Learning to Presolve** (**L2P**). Firstly, we modify simulated annealing to search for the most suitable presolving given each instance. Then, we adapt neural networks that learn the mapping from instance to the found presolving. When applied to unseen instances, the well-trained neural networks can infer suitable presolving in a considerably short time (less than 0.5s in the experiments). Besides, considering the attributes and relations among different presolving parameters, we decide to build hybrid inference networks, in which the priority is regarded as prior knowledge to guide the learning.

We conduct experiments on popular MIP datasets with scales from small to large and two industry-level datasets. Results show that there is indeed much room for improvement in the default presolving of MIP solvers, and the solver can be indeed largely improved when switching the default instance-agnostic presolving into instance-specific presolving by L2P. **This suggests that default presolving is a performance-limiting factor of the solver and deserves more attention.** We consider this task could be a new direction for using machine learning technologies to further improve MIP solvers.

The related works cover different aspects, including solving MIP instances, presolving in MIP solvers, and auto configuration, which we leave in Appendix A.2. The highlights of this paper are four-fold:

**1)** To the best of our knowledge, this is the first work in literature proposing adaptively tailored presolving w.r.t. MIP solvers. Better presolving could significantly reduce the time consumed in solving MIP instances but few works in the literature consider improving it.

**2)** We propose a hybrid neural framework equipped with heuristic algorithms as supervision to predict the suitable presolving for each input MIP instance, which combines the advantage of searching effectiveness of heuristic algorithms and inference efficiency of neural networks.

**3)** Experimental results on both public and private industrial benchmarks show the effectiveness and efficiency of our L2P. It has also demonstrated the necessity of adaptively selecting presolving instead of the default instance-agnostic presolving to boost the performance of the MIP solver.

**4)** We have open-sourced our code as a benchmark of utilizing machine learning to improve the presolving in MIP solvers, please refer to our Github repository for more details.

## 2 FORMULATION AND METHODOLOGY

In this section, first, we introduce presolving and its role in solving MIP. Then, we adopt a simulated annealing based heuristic searching method that can find the most suitable presolving but with large time consumption. Finally, we propose a deep learning approach to learn the presolving found by the simulated annealing with the advantage of efficiency, named learning to presolve (L2P).

### 2.1 PRELIMINARIES: MIP SOLVER AND PRESOLVING

It is well known that any mixed integer linear programming can be written in canonical form:

$$min\{\mathbf{c}^\top \mathbf{x} : \mathbf{Ax} \leq \mathbf{b}, \mathbf{x} \in \mathbb{Z}^p \times \mathbb{R}^{n-p}\}, \tag{1}$$

where $n$ is the number of variables, $m$ is the number of constraints, $\mathbf{c} \in \mathbb{R}^n$ is the objective coefficient vector, $\mathbf{A} \in \mathbb{R}^{m \times n}$ is the constraint coefficient matrix, $\mathbf{b} \in \mathbb{R}^m$ is the constraint right-hand-side vector, and $p \leq n$ is the number of integer variables. We assume the first $p$ variables are integer variables and the last $n - p$ variables are continuous variables.

In general, the branch-and-bound (B&B) algorithm is utilized to solve the MIP instance to global optimal, which follows a divide-and-conquer. However, the B&B algorithm requires lots of time and resources to find the optimal solution if the size of the input MIP instance is large. Therefore, in modern MIP solvers, presolving is conducted to simplify the original instance, as Fig. 1 shows. In the popular open-source MIP solver SCIP, multiple presolvers are used to reduce the size of the model by removing irrelevant information like redundant constraints, strengthening the linear programming relaxation by exploiting integrality information, and extracting useful information in presolving.

There are three key parameters for each presolver: 1) **priority** denotes the order in which different presolvers are executed; 2) **max-rounds** denotes the maximal number of rounds the presolver participates in; 3) **timing** denotes the timing mask of the presolver. In the process of presolving, at every step, one presolver is selected from the presolver pool based on the priority of each presolver. When all presolvers are selected, we refill the presolver pool with the presolvers that are used for less than their max-rounds times. As we can see, the priorities of the presolvers tend to have greater impacts on the performance of presolving, which is also illustrated in (Elble, 2010).

In existing solvers, the parameters of presolvers are all set by default, no matter how the input instance varies. Though the default presolving parameters are designed by experts, we consider using unchanged presolving for changeable inputs is not a good idea. In our opinion, the ideal MIP solver should analyze the feature of the input instance and tailor suitable presolving parameters. In this way, the power of presolving is fully utilized, and so is the power of the whole solver. Therefore, we aim to design a general approach to finding the best presolving parameters given each input instance, in other words, changed from instance-agnostic presolving to instance-specific presolving.

### 2.2 SIMULATED ANNEALING FOR SEARCHING BEST PRESOLVING PARAMETERS

We start with a search-based baseline for tuning the presolving parameters. SCIP includes 14 presolvers and 3 parameters (priority, max-rounds, timing) for each, and one needs to traverse total $42 (14 \times 3)$ parameters which can be challenging. For this purpose, we have tried several popular heuristic tools including Bayesian optimization (BO), simulated annealing (SA), and evolution strategy, we resort to SA (Van Laarhoven & Aarts, 1987) for its suitability for discrete variables which take up 2/3 parameters in our case. While BO with Gaussian processes is designed more suited to continuous variables. Our ablation in Sec. 3.4 shows that SA-based presolving tuning outperforms BO-based one. We place the detail of SA and how we adapted it in our L2P in Appendix (A.3).

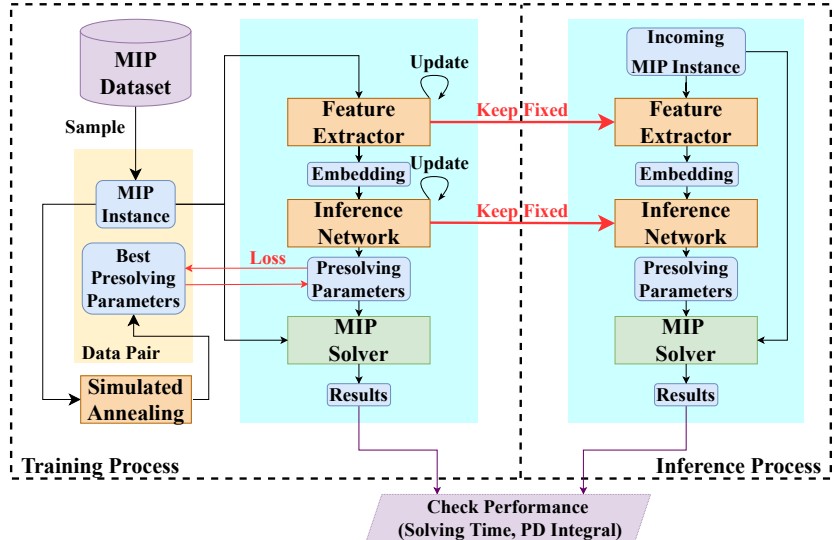

Figure 2: Our proposed framework L2P for learning to presolve. (Left): in the training process, we use simulated annealing to search for the most suitable presolving parameters for each MIP instance, which will be the label to update our neural networks during the network training. (Right): in the inference process, we use the well-trained neural networks from the training process to tailor presolving in an instance-specific manner. Multiple criteria including the solving time and the PD Integral are used to evaluate the performance of the presolving parameters.

## 2.3 L2P: LEARNING TO PRESOLVE

The drawback of SA is its considerable time cost which can be even higher than the time for B&B to actually solve the problem. Therefore, we propose to utilize neural networks along with SA, and the paradigm is termed learning to presolve (**L2P**). We train the neural networks via the data generated by SA in the training set and use the well-trained neural networks for inference when testing. The inference time of neural networks is insignificant and can be readily used in real-world applications.

### 2.3.1 FRAMEWORK DESIGN

As shown in Fig. 2, our proposed L2P includes the training process and inference process. Specifically, for each MIP dataset, we first feed the MIP instances from the training set to the simulated annealing algorithm, which outputs the best presolving parameters for each instance. Then, we regard the data pair (MIP instance / best presolving parameters) as the (input / label) of our neural networks. The input instances first pass our feature extractor and acquire the graph-embedding representation. Then, our inference network takes the embedding and predicts the presolving parameters. Now, for each input MIP instance, we set the predicted presolving to the corresponding position in the MIP solver, and let the modified solver solve the MIP instance. Finally, we analyze the results of the running via multiple criteria including the solving time and primal-dual gap integral. For network updating, we calculate the loss between the best presolving parameters from the simulated annealing (the label) and the predicted presolving parameters from the inference network (the prediction). The loss is passed backward through both the inference network and the feature extractor.

For the inference process, we utilize the well-trained feature extractor and inference network after training. For every incoming MIP instance unseen before, we feed it to the neural networks and acquire the predicted presolving parameters. In the same way, we modify the solver and then solve the instance. Since we save time for simulated annealing, the inference process costs very little time (less than 0.5s) and the whole framework is able to be embedded into real-world MIP solvers.

### 2.3.2 FEATURE EXTRACTOR DESIGN

For the design of the feature extractor, we first represent a given MIP instance as a bipartite graph $(\mathcal{G}, \mathbf{C}, \mathbf{E}, \mathbf{V})$ based on the method in (Gasse et al., 2019). In the bipartite graph, $\mathbf{C} \in \mathbb{R}^{m \times c}$ corresponds to the features of the constraints; $\mathbf{V} \in \mathbb{R}^{n \times d}$ denotes the features of the variables; and an edge $e_{ij} \in \mathbf{E}$ between a constraint node $i$ and a variable node $j$ if the corresponding coefficient $\mathbf{A}_{i,j} \neq 0$.

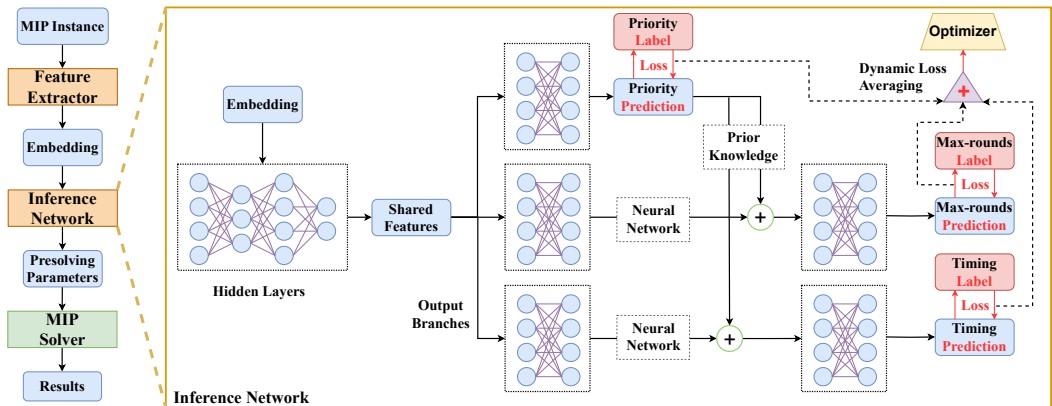

Figure 3: Inspecting the inference network of L2P, follows the data to pass through inside modules. Inspired by knowledge-based residual learning, we regard the priority information as the prior knowledge for learning max-rounds and timing via hybrid neural networks. After calculating three losses, a dynamic loss averaging method is adapted to aggregate them.

We use the same features as the existing work (Prouvost et al., 2020). Next, the bipartite graph is sent as input into a two-interleaved graph convolutional neural network (GCNN) (Gasse et al., 2019). In detail, the graph convolution is broken into two successive passes, one from the variable side to the constraint side, and one from the constraint side to the variable side:

$$
\mathbf{c}_i^{(k+1)} \leftarrow \mathbf{f_C}\left(\mathbf{c}_i^{(k)}, \sum_j^{(i,j)\in\mathbf{E}} \mathbf{g_C}(\mathbf{c}_i^{(k)}, \mathbf{v}_j^{(k)}, e_{ij})\right), \quad \mathbf{v}_j^{(k+1)} \leftarrow \mathbf{f_V}\left(\mathbf{v}_j^{(k)}, \sum_i^{(i,j)\in\mathbf{E}} \mathbf{g_V}(\mathbf{c}_i^{(k)}, \mathbf{v}_j^{(k)}, e_{ij})\right),
$$

(2)

where $\mathbf{f_C}$, $\mathbf{g_C}$, $\mathbf{f_V}$ and $\mathbf{g_V}$ are 2-layer perceptrons. We adopt the ReLU as the activation function. While $k$ represents the number of times that we perform the convolution.

### 2.3.3 INFERENCE NETWORK DESIGN

As for the inference network, we design the shared-bottom neural networks to predict the priority, max-rounds, and timing simultaneously, in other words, make three predictions. As Fig. 3 shows, we first use a hidden layer to process the graph-embedding representation and regard the results as shared features, which are used for all three predictions. As mentioned in Sec. 2.1, we find that priority is the most important among all presolving parameters. Since priority determines the order of all presolvers, it can also balance the influence of all presolvers. In this sense, we believe that priority is more significant than max-rounds and timing. Therefore, we consider designing special hybrid neural networks after the shared features to utilize this property.

As Fig. 3 illustrates, there are three output branches after the shared features, corresponding to priority, max-rounds, and timing respectively. For the priority branch, we use the normal fully connected layers to make predictions. Then, inspired by the knowledge-based residual learning (KRL (Zheng et al., 2021b; Liu et al., 2021)), we consider using the priority as the prior knowledge to better predict the max-rounds and timing. The key idea of KRL is to treat the prior knowledge as a weak learner and use another neural network model to boost it, which turns out to be a hybrid model. In our inference network, the priority is considered as the prior knowledge in the other two branches, and we use two more fully connected layers as the neural network to boost the performance. Due to the page limit, we place the detailed derivation process and proofs of KRL in Appendix (A.4). As proved by KRL, these hybrid knowledge-based residual networks help to reduce the difficulty of learning the parameters and increase the robustness and accuracy of the inference network.

Although the total loss can decrease significantly during the learning process, we observe that the training of the three output branches always converges at different speeds. In fact, it is the hardest task for our inference network to predict adequate priorities for presolvers. The loss of the priority branch can hardly fall as quickly as that of two other branches. Consequently, we have to spend additional time on training the max-rounds and timing branches despite the learning of them having already converged, which could easily lead to over-fitting. To avoid this, we exploit a dynamic loss averaging method (Liu et al., 2019) to respectively assign each output branch a variable weight

Table 1: Performance on easy, medium, and hard datasets. $m$ and $n$ denotes the average number of constraints and variables. We calculate the solving time/PD Integral and report the improvement/effectiveness compared to the default setting. For each instance, **SA runs for hours/days but our L2P only needs milliseconds.** (see more about time difference in Sec. 3.1.4) We run the experiments five times with 5 random seeds and report the average results. (refer to Sec. 3.2)

| | Easy: Set Covering (n = 1000, m = 500) | | | Easy: Max Independent Set (n = 500, m = 1953) | | | Easy: MIRP small (n = 709, m = 841) | | |
|---|---|---|---|---|---|---|---|---|---|
| | Time (s) ↓ | Imprv. ↑ | Effect. ↑ | Time (s) ↓ | Imprv. ↑ | Effect. ↑ | Time (s) ↓ | Imprv. ↑ | Effect. ↑ |
| Default | 7.22 | - | - | 8.66 | - | - | 56.54 | - | - |
| SA | 7.21 | 0.00% | 0% | 8.66 | 0.00% | 0% | 46.67 | 17.46% | 58% |
| Random | 7.27 | 0.00% | 0% | 8.69 | 0.00% | 0% | 54.42 | 3.75% | 21% |
| SMAC3 | 7.24 | 0.00% | 0% | 8.70 | 0.00% | 0% | 50.52 | 10.64% | 42% |
| FBAS | 7.31 | 0.00% | 0% | 8.73 | 0.00% | 0% | 54.64 | 3.44% | 21% |
| **L2P** | 7.22 | 0.00% | 0% | 8.68 | 0.00% | 0% | **50.25** | **11.12%** | **46%** |
| | Medium: Corlat (n = 466, m = 486) | | | Medium: MIK (n = 413, m = 346) | | | Hard: MIRP large (n = 4120, m = 6857) | | |
| | Time (s) ↓ | Imprv. ↑ | Effect. ↑ | Time (s) ↓ | Imprv. ↑ | Effect. ↑ | PD Integral ↓ | Imprv. ↑ | Effect. ↑ |
| Default | 31.02 | - | - | 237.50 | - | - | 2958.83 | - | - |
| SA | 15.93 | 48.63% | 65% | 228.38 | 3.84% | 7% | 1473.75 | 49.81% | 60% |
| Random | 29.34 | 5.43% | 22% | 239.85 | 0.00% | 0% | 2834.26 | 4.21% | 19% |
| SMAC3 | 24.09 | 22.34% | 42% | 233.65 | 1.62% | 6% | 2574.77 | 12.98% | **40%** |
| FBAS | 25.84 | 16.69% | 39% | 238.75 | 0.00% | 0% | 2746.09 | 7.19% | 20% |
| **L2P** | **20.24** | **34.74%** | **55%** | **230.33** | **3.02%** | **7%** | **2118.23** | **28.41%** | 35% |
| | Hard: Item Placement (n = 1083, m = 195) | | | Hard: Load Balancing (n = 61000, m = 64304) | | | Hard: Anonymous (n = 37881, m = 49603) | | |
| | PD Integral ↓ | Imprv. ↑ | Effect. ↑ | PD Integral ↓ | Imprv. ↑ | Effect. ↑ | PD Integral ↓ | Imprv. ↑ | Effect. ↑ |
| Default | 221630.77 | - | - | 5857.95 | - | - | 68319.60 | - | - |
| SA | 210593.56 | 4.98% | 56% | 5550.99 | 5.24% | 36% | 44940.63 | 34.22% | **55%** |
| Random | 221685.75 | 0.00% | 0% | 5879.17 | 0.00% | 0% | 53132.15 | 22.23% | **55%** |
| SMAC3 | 217220.32 | 1.99% | 30% | 5733.76 | 2.12% | 38% | 42460.63 | 37.85% | **55%** |
| FBAS | 222096.19 | 0.00% | 0% | 5862.64 | 0.00% | 0% | 55181.74 | 19.23% | **55%** |
| **L2P** | **210637.88** | **4.96%** | **42%** | **5558.61** | **5.11%** | **48%** | **33278.48** | **51.29%** | **55%** |

when aggregating the three losses. We place the detailed mathematics formulation of dynamic loss averaging in Appendix (A.5). Intuitively, the branch with a slower converge speed would be assigned a larger weight and vice versa. In this way, we can accelerate the learning of priority prediction and thus provide more reliable prior knowledge for the inference of max-rounds and timing.

# 3 EXPERIMENTS

Please note that except for the following subsections, we have placed the additional experiments and discussions in the appendix, including the multiple-run results with standard deviation (A.7), ablation studies by adjusting the size of training data (A.9), experiments on the popular MIPLIB dataset (A.8), illustration of the searching process (A.10), illustration of the improved presolving parameters (A.11), and the discussion of limitations and future work (A.12).

## 3.1 PROTOCOLS

### 3.1.1 DATASETS

We follow (Gasse et al., 2019; 2022) and use popular datasets in our experiments. We evaluate our approach on the four levels of difficulty: easy, medium, hard, and industrial-level datasets:

1) Easy datasets comprise three popular synthetic MIP benchmarks: **Set Covering** (Balas & Ho, 1980), **Maximum Independent Set** (Bergman et al., 2016) and **Maritime Inventory Routing Problem** (MIRP) (Papageorgiou et al., 2014). We artificially generate instances in line with (Gasse et al., 2019; Sun et al., 2021; Jiang & Grossmann, 2015).

2) Medium datasets include **CORLAT** (Gomes et al., 2008) and **MIK** (Atamtürk, 2003), which are widely used benchmarks (He et al., 2014; Nair et al., 2020).

3) Hard datasets from NeurIPS 2021 Competition (Gasse et al., 2022) include **Item Placement**, which involves spreading items that need to be placed; **Load Balancing**, inspired by real-life applications of large-scale systems; **Anonymous**, inspired by a large-scale industrial application; and **Maritime Inventory Routing problem** (MIRP) with hard problem settings.

4) **Private Industrial Benchmarks.** We collect real-world data concerning planning and scheduling in a production planning engine of an influential corporation and formulate them as MIP instances. The production planning problem is to plan daily production for hundreds of factories according to customers' daily and predicted demand. The problem is subject to material transportation and production capacity constraints, which aim to minimize the production cost and lead time simultaneously.

For the datasets used in our experiments, we follow the common usage in existing works (Gasse et al., 2019; Han et al., 2023; Wang et al., 2023; Li et al., 2023a) including splitting data into training and testing sets with 80% and 20% instances. For the easy datasets, we generate 1000 instances for each. For the medium and hard datasets, we directly split the instances provided by their original benchmarks. Here we list these datasets and their total number of instances: Corlat(2000) Gomes et al. (2008), MIK(100) Atamtürk (2003), Item Placement(10000) Gasse et al. (2022), Load Balancing(10000) Gasse et al. (2022), Anonymous(118) Gasse et al. (2022). The two industrial datasets contain 1,000 instances for each, as collected from two periods respectively (from 15 May 2022 to 15 Sept. 2022 and from 8 Oct. 2022 to 8 Dec. 2022).

### 3.1.2 EVALUATION METRICS

Throughout all experiments, we use SCIP 7.0.3 (Gamrath et al., 2020) as the back-end solver, which is the state-of-the-art open-source MIP solver. Note it is nontrivial to test our approach on the commercial solvers e.g. Gurobi, for the limited access to their interfaces. Besides, we use Ecole 0.7.3 (Prouvost et al., 2020) and PySCIPOpt 3.5.0 (Maher et al., 2016) for better implementation. Except for the presolving module, we keep all the other SCIP settings/parameters by default. We use two popular evaluation metrics, i.e., the average solving time (**Time**, lower is better), and the average primal-dual gap integral (**PD integral**, lower is better). To better show the performance of improving presolving, we calculate the improvement (**Imprv.**, the higher the better) made by the compared methods compared to SCIP's default settings. Moreover, we calculate the effectiveness (**Effect.**, the higher the better), *aka.* the "better/win rate". Here "better" means the solving time/PD integral of improved presolving is better than the solving time/PD integral of SCIP's default presolving. The higher effectiveness means more instances the method can find better presolving. The testing is based on the MIP solver itself, and we directly acquire the solving time/PD integral from the solver. The solving time/PD integral contains both the presolving process and the B&B process, which is directly required by SCIP, Ecole, and PySCIPOpt. For details and the mathematics formulation of PD integral, refer to the documentation [1] or our detailed description in Appendix (A.6).

### 3.1.3 IMPLEMENTATION DETAILS

For SA, we set the initial temperature as 1e5 with the decay rate 0.9 until it reaches the minimum temperature of 1e-2. For the neural networks, we use ADAM with a batch size of 32, and learning rate of 1e-4, and a hidden size of 64. For the feature extractor, we follow the same settings as (Gasse et al., 2019) for building graph embeddings. For the hybrid inference networks, we set the hidden size as 64. The loss functions used in our methods are ListMLE (Xia et al., 2008) for the priority and Cross-Entropy (Good, 1952) for the max-round and timing. ListMLE is a loss function designed for ranking, which is suitable for learning the priority since the priority denotes the order/rank of the presolvers. The number of epochs for training is 10,000. The experiments are conducted in a Linux workstation with NVIDIA 3090 GPU and AMD Ryzen Threadripper 3970X 32-Core CPU. Particularly, for the hard datasets in which the instance scale is large, we gradually reduce the batch size to 4 until they can be put into the GPU wholly. Our work can be readily reproduced via these settings and our code in the Github repository.

### 3.1.4 COMPARED METHODS

1) **Default**: following the same presolving parameters by the default setting of SCIP to solve all MIP instances. 2) **Random**: we randomly select the parameters for all presolvers for 10 times and record the best ones. 3) **SMAC3** ((Lindauer et al., 2022)): the latest automatic configuration framework

---

[1] https://www.ecole.ai/2021/ml4co-competition/#metrics

Table 2: Performance on two industrial datasets. $m$ and $n$ denote the average number of constraints and variables respectively. We record the accumulation of the solving time over all instances in the dataset and report the improvement compared to the default setting.

| Industrial Dataset #1 ($n = 1494, m = 5583$) | | | Industrial Dataset #2 ($n = 8456, m = 2392$) | | |
|---|---|---|---|---|---|
| | Solving Time (s) ↓ | Improvement ↑ | | Solving Time (s) ↓ | Improvement ↑ |
| Default | 21280.31 | - | Default | 2994.69 | - |
| SA | 20224.38 | 4.96% | SA | 2347.07 | 21.62% |
| Random | 21387.25 | 0.00% | Random | 2887.32 | 3.57% |
| L2P (ours) | 20420.28 | 4.06% | L2P (ours) | 2447.56 | 18.27% |

aims at finding one single configuration for each MIP category. 4) **FBAS** ((Georges et al., 2018)): one algorithm selection method designed for MIP, which combines several standard ML techniques to select a well-performing algorithm based on a feature description of the input MIP instance. 5) **SA** (simulated annealing): the sub-module in our L2P that uses the simulated annealing to search for the best presolving parameters, of which the time consumption is huge. 6) **L2P** (Ours): our proposed L2P is orthogonal to other progress made in learning for MIP in previous literature.

For every method in our experiments, they use their own algorithm to improve the presolving (running time), and then SCIP uses the improved presolving to solve the MIP instance (solving time), where the solving time/PD integral is used for evaluation. In other words, there are two steps in our experiments: **1) Running step**: we use SA/Random/SMAC3/FBAS/L2P to find a better presolving, and deploy the new presolving in the MIP solver; **2) Solving step** (including presolving and B&B): we use the adjusted MIP solver to solve the MIP instance without further intervention. The metrics (solving time/PD integral) in the tables are directly acquired by the API from SCIP in the solving step.

In the running step, for each instance, our L2P needs less than **0.05s** in the medium datasets, and less than **0.5s** in the hard datasets, while the SA needs hours in the medium datasets and days in the hard datasets. When we claim that the time consumption of SA is unacceptable, we mean its running time. Therefore, we can regard SA as an offline method (running for hours/days) while Random/SMAC3/FBAS/L2P are online methods (running for seconds). The running time of L2P is negligible compared to the improvement it brings. Therefore, we should focus on the comparison among online methods, in other words, between our L2P and Random/SMAC3/FBAS.

## 3.2 EXPERIMENTS ON PUBLIC MIP DATASETS

To verify the performance of our L2P, we conduct experiments on various common MIP datasets in Table 1. We run the experiments with 5 random seeds and report the average improvement and effectiveness compared to the default setting. Due to the page limit, we place the detailed standard deviation results in Appendix (A.7). For the easy datasets, the improvement is not significant. We consider it is because the MIP solver has made sufficient optimization to these classic problems. Besides, the easy datasets are constructed by experts in operational research, in which thus there is not much redundancy. Therefore, even SA cannot find more suitable presolving parameters. **However, it is impossible for the solver to pre-optimize all kinds of MIP instances, especially for real-world MIP instances.** The MIP instances obtained from the real world are constructed by practitioners with different expertise in operational research. Usually, there is much more redundancy in MIP instances from the real world than those from the academy. In the medium datasets, we can see that our proposed methods can make a significant improvement in the Corlat dataset, which can save more than 1/3 of the solving time. When it comes to hard datasets, we change the evaluation metric from solving time to the PD integral, and our L2P still reaches good performances. Compared to SA which needs hours or days to finish searching for each instance, our proposed L2P can make inferences in merely seconds. It turns out that the special design of our proposed L2P for presolving does show its value since L2P can outperform the latest SMAC3 in most cases. Considering that SCIP has been well developed and updated for 20 years, we think our current improvements to SCIP are meaningful for both methodology development and practical use.

## 3.3 EXPERIMENTS ON PRIVATE INDUSTRIAL DATASETS

We conduct experiments on two industrial benchmarks provided by our industrial partner in Table 2. As shown in the caption, the scale of this dataset is large and of relatively high variance. There is still

Table 3: Generalization test on MIRP small and large datasets. We train L2P on the small/large dataset and test it on the large/small dataset respectively.

| Easy:MIRP small ($n = 709, m = 841$) | | | Hard:MIRP large ($n = 4120, m = 6857$) | | |
|---|---|---|---|---|---|
| | Solving Time (s) ↓ | Improvement ↑ | | PD Integral ↓ | Improvement ↑ |
| Default | 56.54 | - | Default | 2958.83 | - |
| Random | 54.42 | 3.75% | Random | 2834.26 | 4.21% |
| L2P (small → small) | 50.25 | 11.12% | L2P (large → large) | 2118.23 | 28.41% |
| L2P (large → small) | 52.66 | 6.86% | L2P (small → large) | 2680.31 | 9.41% |

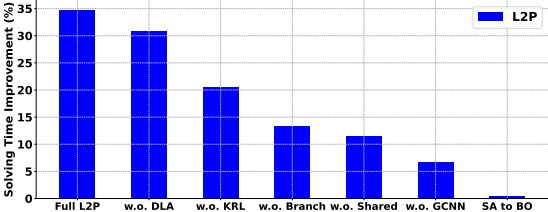

Figure 4: Performance drop by removing components from vanilla L2P on Corlat dataset. We remove the components one by one from the full version of L2P.

Table 4: Generalization test of our L2P w.r.t improvement. We train our L2P on Corlat, MIK, and the combined dataset of Corlat and MIK. Then, we evaluate the performance of L2P on the original Corlat and MIK dataset.

| train
test | Corlat | MIK | Corlat + MIK |
|---|---|---|---|
| Corlat | **34.74%** | 15.52% | 33.84% |
| MIK | 1.12% | **3.02%** | 1.87% |

a large improvement room for the default setting. SA can find more suitable presolving parameters, which reduces the solving time by $4.96\%$ and $21.62\%$ on the two datasets, respectively. Our method still shows a notable improvement compared to the default setting and the performance gain ($4.06\%$ and $18.27\%$) is close to SA's, while SA needs hours to run but our L2P only needs seconds. Due to potential privacy issue, we did not test SMAC3 and FBAS on these datasets.

## 3.4 GENERALIZATION TEST AND ABLATION STUDY

To test the generalization of L2P, first, we conduct experiments on MIRP small and large datasets. The two datasets are significantly different in complexity denoted by scale. We train L2P with MIRP small dataset and observe its performance on MIRP large datasets, and vice versa. The statistics of the datasets and experiment results are reported in Table 3. We can see that on both tests our L2P outperforms the baselines when generalized from another dataset, which denotes its generalization effectiveness. Besides, we add more experiments in Table 4, where the experimental settings are the same as the settings in Table 1, including the data size and the metric. Instead of training domain by domain normally, we try multiple train/test settings on different domains. In the first two columns, we can see that L2P can handle unseen domains (families) of instances. In the last column, we train L2P with the mixed dataset and L2P still reaches good performance when testing on both two domains.

Fig. 4 shows the effect of removing components in L2P one by one: from the dynamic loss averaging (DLA), to the hybrid knowledge-based residual learning (KRL), to the output branches, to the shared feature, to GCNN, and at last, we replace SA with Bayesian optimization. We can see that removing the KRL module leads to a nearly 10% performance drop. Therefore, the hybrid KRL structure in L2P is more significant. When we change SA to BO, we note that BO can hardly find better presolving parameters, as the improvement downgrades from 10% to almost 0%.

## 4 CONCLUSION AND OUTLOOK

We propose a paradigm of learning to presolve for MIP solvers. Instead of using the default instance-agnostic presolving as in existing solvers, we use simulated annealing to search for suitable presolving in an instance-specific manner. Furthermore, we design the hybrid neural networks to learn the results generated from SA. Experiments on both public and private MIP datasets show its performance and cost-efficiency. We hope our results and open-source can draw wide attention and further evaluation could be performed on commercial solvers which we believe is a less-studied yet promising area. One possible future work is to combine the learning-based solvers e.g. (Zhang et al., 2024) tailored to MIP and more general methods (Li et al., 2023c) with our presolve techniques. Also, one may combine other instance generation models (Li et al., 2023b; Chen et al., 2024) for training set augmentation. For more discussion of potential limitations and future plans, please refer to Appendix (A.12).

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

# A   APPENDIX

## A.1   PRESOLVERS IN MIP SOLVERS

Seeing its important role, different presolvers have been devised to modify the MIP instance before actually solving it. In the most popular open-source solver SCIP Gamrath et al. (2020), 14 presolvers are used in presolving. In Table 5, we list these presolvers with brief descriptions. For more information, please refer to the official document of SCIP [2].

Table 5: Brief descriptions of fourteen commonly used presolvers in SCIP. For CPLEX the cases are similar, while for Gurobi we can not acquire their interfaces since Gurobi is a commercial solver and not open-source.

| Presolver | Description |
| --- | --- |
| Boundshift | converts domain [a,b] to domain [0,b-a] |
| Convertinttobin | converts integer variables with domain [a, a+1] to binaries |
| Domcol | finds dominance relations between variables and derives lower bound |
| Dualagg | aggregates the variables with specific structure |
| Dualcomp | fixes the bound if the combination of continuous variables can compensate |
| Dualinfer | strengthens the bounds on continuous variables |
| Gateextraction | extracts gate-constraints constraints and set-partitioning constraints |
| Implics | implication graph presolver which checks for aggregations |
| Inttobinary | converts integer variables with domain [a,a+1] to binaries |
| Milp | calls the presolve library and uses the postsolve information |
| Qpkktref | tries to add the KKT conditions as additional constraints |
| Redvub | removes redundant variable upper bound constraints |
| Stuffing | investigates singleton continuous variables if can be fixed at a bound |
| Trivial | fixes variables with equal bounds to this value |

## A.2   RELATED WORK

In this subsection, we discuss the existing works closely related to ours: 1) solving MIP instances, which is the problem we focus on; 2) presolving and presolvers, which is the insight of this work; 3) Automatic configuration and algorithm selection as our work roughly falls into this category; 4) machine learning for combinatorial optimization, which is a general approach of combining machine learning into traditional solvers.

**Solving MIP Instances.** Machine learning methods have shown the potential to accelerate the solving of MIP instances. Specifically, many works learn to select branching variables (Etheve et al., 2020; Qu et al., 2022; Gupta et al., 2022) or parameterizing B&B search tree directly (Zarpellon et al., 2021; Lin et al., 2022). Moreover, node selection is also an appealing scenario where imitating learning can be leveraged to speed up the solving process (He et al., 2014; Yilmaz & Yorke-Smith, 2020). Due to the significance of cutting planes to B&B, ML techniques have also been applied to train cut selection policy to improve the tightening of feasible region (Huang et al., 2022; Paulus et al., 2022; Berthold et al., 2022). One recent work (Kuang et al., 2023) aims at using RL to improve the solving of linear programming. However, rare works investigate how another important module in MIP solvers, i.e., the presolving module, can benefit from learning-based technologies. We recommend these surveys for more details (Li et al., 2024; Zhang et al., 2023).

**Presolving in MIP Solvers.** Presolving plays a key role in MIP, which can improve the model constraints' description of the underlying polyhedron of integer-feasible solutions (Achterberg et al.,

---

[2]https://scipopt.org/doc/html/group__PRESOLVERS.php

2019). Several studies (Bixby & Rothberg, 2007; Achterberg & Wunderling, 2013a) have empirically explored how various components in the MIP solvers affect the solution quality and one consensus is that presolving is one of the most powerful components. Essentially, presolving can be considered a set of methods to drop redundant information of model formulation (Brearley et al., 1975), such as removing redundant constraints, fixing variables, and seeking generalized upper bounds. (Achterberg et al., 2019) introduced the presolving techniques as used in Gurobi. Despite the significance of presolving, there is surprisingly relatively limited literature for comprehensive study, especially from the machine learning perspective for introducing adaptiveness for presolving.

**Automatic Configuration and Algorithm Selection.** Automatic configuration tries to find a single robust configuration across a set of problem instances. ParamILS (Hutter et al., 2009) provides a complete framework for parameter tuning and algorithm configuration. (Hutter et al., 2011) further uses Bayesian optimization to find better configurations. The open-source SMAC3 (Lindauer et al., 2022) is a robust and flexible framework for BO in determining well-performing hyperparameter configurations. Algorithm selection (Kotthoff, 2016; Kerschke et al., 2019) aims at choosing among a set of algorithms the ones that are likely to perform best for a particular instance. (Georges et al., 2018) combine several standard ML techniques to select a well-performing algorithm based on a feature description of input MIP instances. Instead of finding a single configuration for all instances, we tend to design an instance-specific presolving tailor method. Besides, our L2P focuses on the presolving module with consideration of presolver order, which is hardly studied in the literature.

**Learning for Combinatorial Optimization.** There are growing interests in using learning in solving combinatorial optimization problems (Bengio et al., 2021), researchers have considered adopting deep learning in several NP-hard problems, such as traveling salesman problem (Li et al., 2023c; Khalil et al., 2017), vehicle routing problem (Nazari et al., 2018; Zheng et al., 2021a), job scheduling problem (Chen & Tian, 2019), maximal common subgraph (Bai et al., 2021), graph matching (Liu et al., 2023; 2022b; Wang et al., 2024), model fusion (Liu et al., 2022a). Aside from directly leveraging learning to problem solving, methods are also developed to provide support to solving, such as utilizing generative modeling (Li et al., 2022; 2023d) to augment real-world combinatorial instances (Li et al., 2023b; Chen et al., 2024) and evaluating the robustness of combinatorial solvers (Lu et al., 2023). As a classic problem in combinatorial optimization, researchers have tried to utilize the power of deep learning to accelerate the solving of MIP during B&B (Gasse et al., 2019). However, no works focus on the presolving part to the best of our knowledge.

## A.3   DETAILS OF UTILIZING SIMULATED ANNEALING IN L2P

To find good solutions, SA continuously perturbs the known solution. If the new solution has a better objective value, it is accepted. Otherwise, it could be accepted by a certain probability, which can be specified as: This mechanism helps SA jump out of the possible local optima. Generally, an exponential function is used to set the probability of accepting the worse solution:

$$P = e^{\frac{-\Delta y}{T}} \tag{3}$$

where $T$ denotes the current temperature and $\Delta y$ represents the difference between the objective value of the new and old solution. In the task of this paper, the objective value refers to the solving time/PD integral of solving MIP instances. By continuously lowering the temperature, SA simulates the material coming into equilibrium, and thus effectively simulates the physical annealing. In practice, SA starts initially with $T$ set to a high value, and then it is decreased at each step following a specific schedule, and SA terminates when the temperature $T$ reaches a preset threshold. In this way, the algorithm is supposed to wander initially towards a broad region of the search space containing good solutions and gradually drift towards better regions that become narrower and narrower, and finally move downhill according to the steepest descent heuristic.

When adopting SA to our presolving scenario, we set the first 14 variables as continuous variables and the last 28 variables as integer variables, corresponding to priority (float), max-rounds (int), and timing (int) respectively. The difference between continuous variables and integer variables is the disturbance (for the integer variables we only use $\pm 1$). As for the object function, the most valuable one is the solving time. However, when facing large-scale instances, we can not use the solving time as the objective function if the solving time is clearly more than 3600 seconds. Therefore, instead of solving the MIP instance to the optimal, we set a time limit and output the current solution.

Based on the current solution, we design three surrogate objective functions: 1) int ratio: count the number of variables that are integers and should be integers by the MIP constraints. 2) bound gap: calculate the gap between the dual bound and primal bound as the bound gap. 3) primal-dual gap integral: the area between the curve of the solver's primal bound and dual bound. In our experiments, we find that all three surrogate objective functions work well. For each dataset, we select the surrogate objective function with the best performance in SA.

## A.4 DETAILS OF KNOWLEDGE-BASED RESIDUAL LEARNING

Inspired by (Zheng et al., 2021b), we decide to utilize the knowledge-based residual learning (KRL) in our neural networks, regarding the priority information as the prior knowledge. Here we describe the detailed design of KRL. As Fig. 5 shows, KRL is a residual learning method to combine the prior knowledge model and the neural network model. Specifically, KRL utilizes a simple but effective residual unit with domain knowledge:

$$\mathbf{x}' = \rho(\mathbf{x}) + G_\phi(\mathbf{x}) \tag{4}$$

where $\rho(\mathbf{x})$ is the prior knowledge model, $G_\phi(\mathbf{x})$ is a neural net model. Intuitively, the prior knowledge model can help to infer reasonably well. Hence, the neural network model will be directed to predict the residual $\mathbf{x}' - \rho(\mathbf{x})$. As (Shamir, 2018) shows, learning the residual value $\mathbf{x}' - \rho(\mathbf{x})$ is provable better than learning the original value $\mathbf{x}$ when $G_\phi$ satisfies certain criteria and the output layer is linear. This ensemble significantly increases its robustness and accuracy. Moreover, theoretically, it is guaranteed to yield superior performance over either a pure domain knowledge model or a pure deep learning model. (proved in Section 4.3 of the KRL paper (Zheng et al., 2021b))

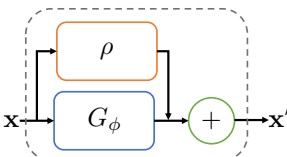

Figure 5: The knowledge-based residual learning unit. $G_\phi$: neural layers, $\rho$: prior knowledge model. Credit to (Zheng et al., 2021b).

## A.5 DETAILS OF DYNAMIC LOSS AVERAGING.

Imbalance among multiple tasks is one of the most critical issues to be addressed in multi-task learning, many studies have attempted to balance the convergence rate of different tasks by assigning an adaptive weight to each task (Kendall et al., 2017; Chen et al., 2017; Liu et al., 2019). Inspired by them, the motivation of our Dynamic Loss Average method is to average task weighting over time via adjusting the rate of change of loss for each task. Formally, we define a loss weight: $\omega_k$ for each task: $k$ as:

$$\omega_k(t) := \frac{K \, exp(\mathcal{R}_k(t-1)/\mathcal{T})}{\sum_i exp(\mathcal{R}_i(t-1)/\mathcal{T})}, \tag{5}$$

$$\mathcal{R}_k(t-1) = \frac{\mathcal{L}_k(t-1)}{\mathcal{L}_k(t-2)} \tag{6}$$

where $\mathcal{L}_k(t)$ denotes the loss of task $k$ at the $t$ iteration and $\mathcal{R}_k(\cdot)$ calculates the relative training convergence rate of task $k$, $\mathcal{T}$ in the softmax operator represents the temperature that controls the softness of weighting for task-specific loss. In the implementation, temperature $\mathcal{T}$ is set to 2 and $\mathcal{R}_k(t)$ is initialized as 1 when $t = 1, 2$, but any other effective initialization methods with prior knowledge can also be applied here.

## A.6 DETAILS OF PD INTEGRAL

For the definition of PD integral, the primal-dual integral in the field of mixed integer programming (MIP) is a measure used to evaluate the performance of MIP solvers. Here we list several documenta-

Table 6: Performance on easy, medium, and hard datasets. We ran the experiments five times with 5 random seeds and reported the average results with their standard deviations.

|  | MIRP small | Corlat | MIK | MIRP large | Item Placement | Load Balancing | Anonymous |
|---|---|---|---|---|---|---|---|
| Random | 3.75% | 5.43% | <0 | 4.21% | <0 | <0 | 22.23% |
| SMAC3 | 10.64%±1.37% | 22.34%±1.08% | 1.62%±0.29% | 12.98%±1.68% | 1.99%±0.11% | 2.12%±0.09% | 37.85%±0.99% |
| FBAS | 3.44%±1.87% | 16.69%±2.26% | <0 | 7.19%±2.43% | <0 | <0 | 19.23%±2.95% |
| L2P (ours) | 11.12%±1.56% | 34.74%±1.57% | 3.02%±0.28% | 28.41%±1.62% | 4.96%±0.11% | 5.11%±0.21% | 51.29%±1.55% |

tion [3] [4] of implementing PD integral for understanding it. It provides a quantitative assessment of how effectively a solver bridges the gap between the primal (the best feasible solution found) and dual (a bound on the optimal solution) bounds over the course of the solution process. Here we list a more detailed breakdown:

- Primal and Dual Solutions in MIP: In mixed integer programming, a primal solution refers to a feasible solution that satisfies all the constraints of the MIP model. The dual solution, on the other hand, is related to the bounds on the optimal solution value. In linear programming, this would be equivalent to the solution of the dual problem, but in MIP, it usually refers to a bound derived from a relaxation of the integer constraints.

- Gap Measurement: The primal-dual integral measures how the gap between the primal and dual solutions evolves over time. This gap is the difference between the objective value of the best known feasible solution (primal) and the best known bound (dual).

- Purpose of the Primal-Dual Integral: This measure is especially useful in understanding the efficiency of a solver in converging to the optimal solution. It can highlight whether a solver quickly identifies good feasible solutions and tight bounds or whether it struggles to improve the primal and dual solutions over time.

- Calculation: The primal-dual integral is calculated by integrating the gap over the time taken for the computation. A lower integral value indicates a more effective solver, as it shows that the solver was able to keep the primal-dual gap smaller throughout its run.

- Use in Analysis and Benchmarking: This metric is valuable for comparing different solvers or solution strategies in mixed integer programming. It provides insights beyond just the final solution quality or computation time, focusing on the overall trajectory of the solution process.

### A.7 EXPERIMENTS WITH 5 RANDOM SEEDS

We conduct experiments with 5 random seeds, following the settings in (Gasse et al., 2019). For the baseline Random, its approach is already randomly selecting the parameters 10 times and recording the best ones. Therefore, we do not modify this baseline. Table 6 shows the solving time/PD integral improvement of the methods on all datasets, the average results with standard deviation are reported. We can see that our proposed methods can make significant improvements on various datasets. Compared to SA which needs hours or days to finish searching for each instance, our proposed L2P can make inferences in merely seconds (less than 0.05s in the medium datasets, and less than 0.5s in the hard datasets). As for the baselines, we find that SMAC3 reaches comparable performance while FBAS does not work well except for Corlat and MIK. It turns out that the special design of our proposed L2P for presolving does show its value since L2P can outperform the latest SMAC3 in most cases, including instance-specific tailor, hybrid residual networks, dynamic loss, and shared-bottom branching. As we can see, in these popular MIP datasets, our L2P has shown its performance and cost efficiency.

---

[3] https://www.ecole.ai/2021/ml4co-competition/#metrics
[4] https://www.fico.com/fico-xpress-optimization/docs/latest/solver/optimizer/HTML/PRIMALDUALINTEGRAL.html

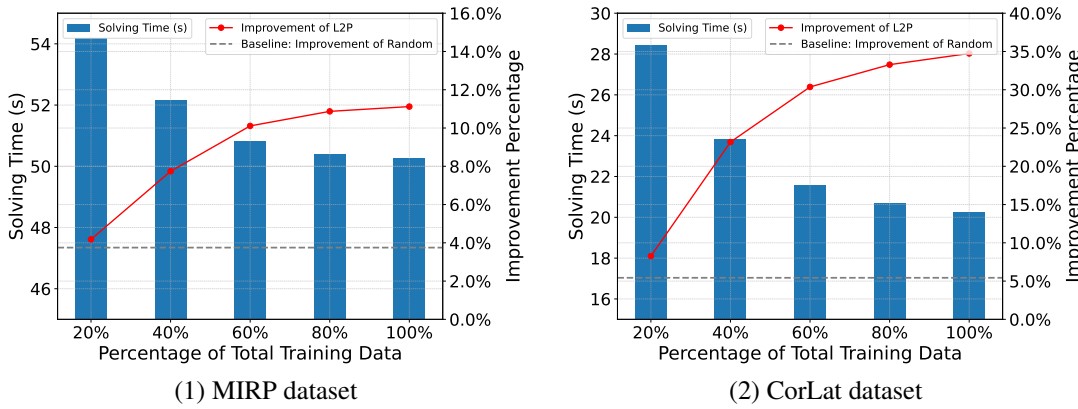

(1) MIRP dataset  (2) CorLat dataset

Figure 6: Ablation study: impact of training data size on the performance of L2P in the MIRP and CorLat dataset.

## A.8   Experiments on the MIPLIB dataset

In the field of MIP, MIPLIB is the most widely used dataset, and we have thought of this dataset during our work period. However, the MIPLIB dataset contains diverse instances which are too diverse to be regarded as one distribution, as in the machine learning dataset we require the data from an independent identically distribution. Known from their website[5], they try to select instances as diverse as possible with multi-round filtering and clustering. It makes the MIPLIB dataset very hard to become a good dataset with independent identical distribution.

Nevertheless, we still tried to search for better presolving than default on MIPLIB via heuristics, but we ended up finding there is little improvement on this benchmark. Since we can not find enough labels (better presolving) for constructing the dataset, we did not include MIPLIB in our paper. We plot the search progress below: (represented by the average improvement compared to the default)

Table 7: Searching attempts on the MIPLIB dataset.

| Searching step | 0 | 15 | 30 | 45 | 60 | 75 | 90 | 105 |
|---|---|---|---|---|---|---|---|---|
| Average improvement | -89.6% | -70.0% | -31.6% | -34.0% | -68.1% | -12.2% | -4.31% | -3.84% |
| Searching step | 120 | 135 | 150 | 165 | 180 | 195 | 210 | 225 |
| Average improvement | -3.26% | -0.18% | -0.09% | 0.04% | 0.69% | -0.75% | -4.34% | -8.29% |

Our analysis is as follows: as a standing benchmark for evaluating MIP solvers themselves, instances of MIPLIB may be well pre-processed such that default presolving is already improved well by the researchers of the MIP solvers, including manually adding new presolvers and adjusting their default strategies. Therefore, there is little room for us to further improve the presolving in MIPLIB. While in the real world, it is impossible to let researchers manually pre-improve with every incoming instance. Therefore, our method can still make improvements to MIP solvers, as the experiments on multiple public datasets show.

## A.9   Ablation study of adjusting the training data

To see the influence of the training data, we conduct an ablation study by adjusting the number of training data, from 20%, 40%, 60%, 80%, to 100%. The results are shown in Fig. 6. We can see that with more training data used, L2P will perform better. But we also notice that after using 80% of the data, the performance of L2P tends to stabilize. Therefore, the idea of finding a suitable data size for training indeed makes sense, and we will keep following.

---

[5] https://miplib.zib.de/Selection_Methodology.html

Table 8: The improvement (%) compared to default presolving from the searching process of SA.

| instance | step 0 | step 15 | step 30 | step 45 | step 60 | step 75 | step 90 | step 105 | step 120 | step 135 |
|----------|--------|---------|---------|---------|---------|---------|---------|----------|----------|----------|
| avg. | -15.7 | 10.4 | 16.4 | 17.5 | 19.0 | 23.0 | 28.3 | 35.3 | 38.1 | 39.2 |
| 1 | -26.4 | -2.1 | -2.1 | 9.1 | 9.1 | 9.1 | 24.0 | 24.0 | 24.0 | 30.3 |
| 2 | -21.1 | 17.2 | 17.2 | 17.2 | 17.2 | 17.2 | 17.2 | 43.9 | 45.9 | 46.1 |
| 3 | -20.8 | 6.2 | 6.2 | 6.2 | 14.3 | 14.3 | 22.1 | 22.1 | 39.4 | 39.4 |
| 4 | -17.7 | -10.8 | -8.9 | -8.9 | -8.9 | 17.6 | 36.1 | 36.1 | 36.1 | 36.1 |
| 5 | -17.3 | 3.2 | 20.8 | 20.8 | 20.8 | 20.8 | 31.2 | 31.2 | 31.3 | 33.1 |
| 6 | -15.5 | 29.3 | 29.7 | 29.7 | 29.7 | 29.7 | 29.7 | 29.7 | 29.7 | 30.6 |
| 7 | -14.0 | 25.0 | 25.0 | 25.0 | 25.0 | 25.0 | 25.0 | 39.2 | 43.3 | 45.5 |
| 8 | -10.8 | -2.1 | 2.8 | 2.9 | 9.9 | 9.9 | 9.9 | 34.6 | 38.4 | 38.4 |
| 9 | -8.4 | 8.2 | 41.8 | 41.8 | 41.8 | 41.8 | 41.8 | 41.8 | 41.8 | 41.8 |
| 10 | -5.6 | 29.7 | 31.2 | 31.2 | 31.2 | 44.6 | 46.2 | 51.0 | 51.0 | 51.0 |

## A.10 ILLUSTRATION OF THE SEARCHING PROCESS

To show the information about the generated training set, we plan to show the searching process of the simulated annealing module of the proposed L2P. As Table 8 shows, we can see that SA can indeed search for better presolving, and its convergence seems fine. For the time consumption of each step, the time limit is 3600s. As a result, we find that SA can indeed find better presolving during the tremendous searching process, but the time consumption of it is unacceptable in practice usage since the time consumption of SA is even larger than the B&B algorithm. Therefore, we propose L2P to learn the results of SA while saving running time. Compared to SA which needs hours or days to finish searching for each instance, our proposed L2P can make inferences in merely seconds (less than 0.05s in the medium datasets, and less than 0.5s in the hard datasets).

## A.11 ILLUSTRATION OF IMPROVED PRESOLVING PARAMETERS

In this paper, we propose the learning to presolve task, which aims at improving the presolving module in existing MIP solvers. To achieve this goal, we propose our L2P to improve the presolving parameters. Here, we consider an illustration of the improved presolving parameters that can better reflect the performance of our proposed method. However, the presolve may not be easy to quantify due to its complexity. In Fig. 7 8 9, we try to show the illustrations for improved presolving parameters of 14 different presolvers under the CorLat, MIK, and NeurIPS 21 Anonymous datasets, in Figure 1/2/3, respectively. Due to the space limit, we choose the top four instances for each dataset by their ranking in the original public zip (not randomly picked). In each figure, we can see that though these four instances are from the same problem classes, the parameters of the improved presolving vary a lot, especially with respect to the priority (order of the presolvers). Meanwhile, we can find that different problem classes show different patterns: some approximately wave-like patterns in each figure. Moreover, we use Tabel 9 to show the decisions of the trained model L2P of the top 10 instances in the CorLat datasets. The [x,y,z] denotes the priority, max-round, and timing of each presolver (decided by our L2P). We also place the default parameters of the presolvers to compare.

## A.12 DISCUSSION OF LIMITATIONS AND FUTURE WORK

In this paper, we focus on the mixed integer programming (MIP) area and propose the "learning to presolve" task for the first time. We design a deep learning framework integrated with heuristic algorithms, which achieves non-trivial performance on multiple public MIP datasets. We hope our work could lead to a new direction in machine learning for combinatorial optimization (ML4CO) by proposing a new task definition and becoming a baseline for other researchers.

Here, we try to list some interesting topics which may be a new direction for future work:

1) Extension to non-linear MIP: our method is designed as a plug-in to the MIP solver, focusing only on the presolve part. As the presolving part is algorithmically agnostic to the blackbox of the MIP

solver itself and the MIP instance, we can work on the MIP instance as long as the MIP solver can work on it. We plan to try some experiments about non-linear MIP in future work.

2) About scaling and decomposition: taking the scheduling problem as an example, when the problem size increases by 10 times, the solving time usually increases by more than 10 times; unless it can be strictly decomposed into subproblems (DoComponents), it can achieve the speed difference of only 10 times mentioned here; generally speaking, even if the problem size of the same scenario increases by 10 times, due to the coupling of the problem, the presolve module will also increase more than 10 times the workload. Taking the common dominatedCols, dominatedRows, multiRowBoundStr in MIP as an example, the judgment method is also $O(n^2)$ and above difficulty. In practice, we have also encountered the situation of high time consumption of presolve operators caused by coupling problems, and the time consumption bottleneck lies in SymmetryDetection. It is very necessary to adjust the presolve parameters according to the scenario. Meanwhile, if some problems have strong decomposition and solving properties (such as continuous scheduling problems, multi-stage planning problems, etc.), we can also adjust and train the presolving according to the problem scenarios after decomposition, which shows the general applicability of our method: not only suitable for larger scenarios but also provide value for decomposable scenarios, which is also a future work direction.

3) Real-world scenarios to use our method: In practice, it is common to repeatedly solve similar MIPs collected from specific real-world applications, e.g., day-to-day production planning and vehicle routing problems. These problems are constructed as mathematical programming models (here, we refer to MIP) by human experts in business scenario understanding and operation research. However, there might be a bunch of redundancy in these MIP models. Presolving techniques play a key role in removing redundancy and enhancing the numerical representation of the MIP model. Many modern MIP solvers such as Gurobi and SCIP are equipped with kinds of advanced presolving operators, which are hard-coded heuristics designed by experts. However, hard-coded presolving does not take into account underlying patterns changing among these problems (even if these problems originate from the same business scenario). Thus, we propose our method to exploit the existing presolving operators better, to solve those similar MIPs faster and more stably.

4) Usage on more MIP solvers other than SCIP: commercial MIP solvers may not have the APIs like SCIP, but as long as the presolving module exists, they may have similar parameters to adjust the presolvers. Their APIs are not exposed since they are commercial solvers and not open-source. We consider it impossible to directly improve commercial solvers in research papers, as existing papers mainly conduct experiments on open-source solvers (SCIP) like us. If we cooperate with the commercial solver company in the future, we can also add our method to their internal strategy, integrating our machine learning technologies to help users better solve MIPs.

People may think that we can write the presolved MIP to disk and then run it on the commercial solvers. We have considered this approach before. Presolving using an open-source MIP solver and then solving the presolved instance using a commercial solver is indeed a feasible approach. Still, we are concerned that this may raise potential issues, as it requires disabling the presolving module of the commercial solver. Considering that the presolving module and the subsequent model solving are not fully decoupled, presolving algorithms may still be executed when solving sub-mips on non-root nodes and enabling the restart mechanism. If we hastily turn off the presolving module of the proprietary commercial solver, it may have unpredictable impacts on its performance. Moreover, the presolving strategy obtained by training on the open-source solver may not be applicable to all solvers. It may also result in negative optimization (we did not verify this point due to time and paper topic constraints). Therefore, we choose a more open SCIP solver to provide personalized presolving settings and complete solving services.

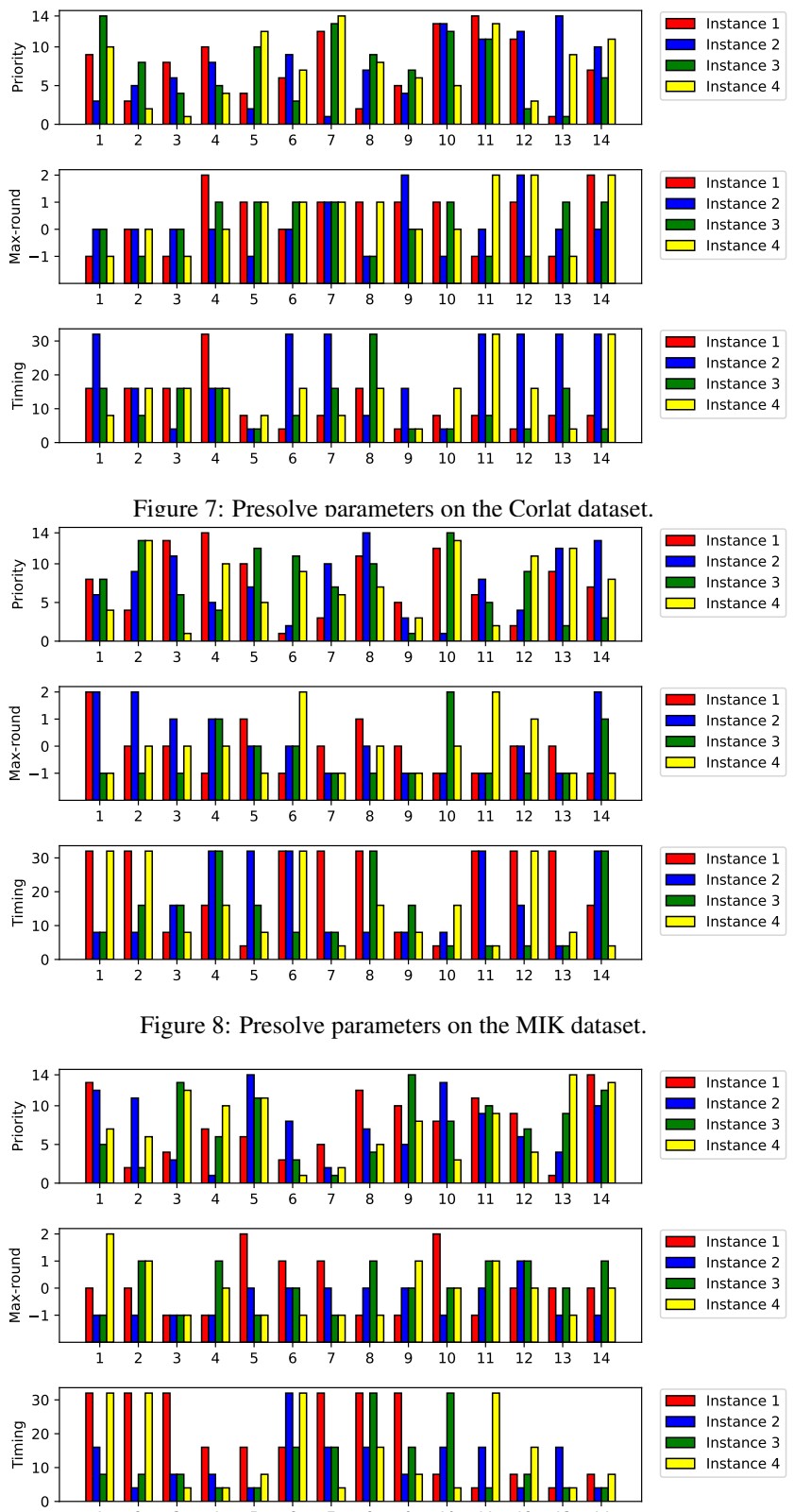

Figure 7: Presolve parameters on the Corlat dataset.

Figure 8: Presolve parameters on the MIK dataset.

Figure 9: Presolve parameters on the NeurIPS 2021 competition anonymous dataset.

Table 9: Detailed presolve parameters for each presolver on the Corlat dataset

| | presolver 1 | presolver 2 | presolver 3 | presolver 4 | presolver 5 | presolver 6 | presolver 7 | presolver 8 | presolver 9 | presolver 10 | presolver 11 | presolver 12 | presolver 13 | presolver 14 |
|---|---|---|---|---|---|---|---|---|---|---|---|---|---|---|
| Default | [13, 0, 4] | [11, 0, 4] | [7, -1, 16] | [3, 0, 16] | [8, -1, 16] | [5, -1, 16] | [10, -1, 16] | [4, -1, 8] | [12, -1, 4] | [9, 0, 8] | [1, 0, 16] | [14, -1, 4] | [6, 0, 16] | [2, -1, 16] |
| Instance 1 | [7, 0, 16] | [13, -1, 4] | [3, 0, 8] | [12, 0, 4] | [4, 1, 16] | [6, 0, 32] | [14, 0, 32] | [1, 0, 32] | [5, 0, 4] | [11, -1, 8] | [9, 2, 8] | [8, -1, 4] | [2, -1, 16] | [10, 0, 8] |
| Instance 2 | [5, -1, 16] | [2, 0, 16] | [10, -1, 16] | [11, -1, 32] | [3, -1, 8] | [1, 1, 16] | [12, 0, 32] | [4, 0, 4] | [6, 1, 4] | [14, 0, 16] | [9, -1, 16] | [8, 1, 4] | [7, -1, 8] | [13, 2, 32] |
| Instance 3 | [5, -1, 32] | [2, 0, 8] | [14, 1, 8] | [13, 0, 32] | [1, 1, 16] | [3, 0, 32] | [11, -1, 32] | [4, 0, 4] | [6, -1, 4] | [12, 0, 16] | [10, -1, 8] | [9, -1, 4] | [7, 1, 32] | [8, -1, 32] |
| Instance 4 | [4, 0, 8] | [6, 1, 4] | [1, 1, 32] | [8, 0, 32] | [7, 1, 4] | [13, -1, 32] | [14, 0, 4] | [12, 1, 8] | [9, -1, 8] | [11, -1, 32] | [3, 1, 4] | [5, 1, 32] | [10, 1, 8] | [2, 0, 4] |
| Instance 5 | [3, -1, 16] | [8, 2, 8] | [12, 0, 32] | [4, 0, 4] | [6, 0, 4] | [10, 0, 16] | [5, 2, 32] | [7, -1, 8] | [13, -1, 32] | [1, 0, 8] | [9, 2, 32] | [11, 2, 32] | [14, 0, 4] | [2, 1, 4] |
| Instance 6 | [2, 1, 8] | [9, 0, 4] | [1, 1, 8] | [11, 0, 4] | [10, 2, 32] | [8, 1, 32] | [14, 0, 32] | [5, -1, 8] | [7, 0, 16] | [13, -1, 32] | [3, 1, 8] | [4, 1, 8] | [6, 0, 4] | [12, 0, 4] |
| Instance 7 | [11, 2, 8] | [14, -1, 4] | [8, -1, 4] | [13, 0, 8] | [6, 0, 16] | [3, 1, 4] | [10, 1, 32] | [12, -1, 8] | [7, 1, 16] | [9, -1, 8] | [5, -1, 32] | [2, -1, 8] | [1, -1, 32] | [4, 0, 4] |
| Instance 8 | [13, -1, 4] | [3, 1, 32] | [6, 2, 32] | [2, 1, 8] | [5, -1, 4] | [12, -1, 4] | [8, 0, 4] | [7, 1, 8] | [11, -1, 8] | [10, 1, 32] | [9, -1, 32] | [4, 1, 32] | [14, 2, 16] | [1, -1, 16] |
| Instance 9 | [2, 0, 4] | [14, -1, 32] | [6, 0, 16] | [9, -1, 4] | [3, 0, 4] | [4, 1, 4] | [12, 0, 4] | [1, 0, 8] | [11, -1, 4] | [7, 0, 32] | [13, 0, 4] | [5, -1, 32] | [8, 1, 16] | [10, 1, 16] |
| Instance 10 | [12, 2, 8] | [1, 1, 8] | [3, 0, 32] | [2, 1, 32] | [14, 1, 32] | [9, 0, 32] | [4, -1, 32] | [8, -1, 16] | [11, 0, 32] | [10, -1, 8] | [6, -1, 32] | [7, 1, 16] | [13, 0, 8] | [5, 1, 8] |

