# OpenReview forum: "L2P-MIP: Learning to Presolve for Mixed Integer Programming"
_ICLR.cc/2024/Conference — ICLR 2024 poster_

### Official Review · Reviewer_hVoV · 2023-10-27

**Soundness:** 3 good
**Presentation:** 4 excellent
**Contribution:** 3 good
**Rating:** 8
**Confidence:** 3

**Summary:**

MIP solvers make use of various presolving methods to improve their performance, and the parameters for these methods can be tuned for the specific problem instance for maximal effectiveness. The authors propose a deep neural net model that will select those parameters by analyzing the MIP problem structure. This neural network is trained using the solutions found from simulated annealing (SA). The SA takes too much running time to be used for practical cases, but the trained neural net, whose output will mimic the result of the SA, can get the job done very fast, allowing for both good speed and good performance.

**Strengths:**

This is an excellent pioneering work on applying a learning approach to the presolving stage of MIP solvers. The results are strong, and the proposed method may be readily applied to practical cases.

**Weaknesses:**

Reports on the details of the neural net training procedures are missing. For example, what are the sizes of the training set/test set? How many epochs were used for training? How much time used? How long did it take to prepare the SA solutions in each case?

Perhaps the most important aspect of this method in terms of practical usefulness is the number of training data needed for the neural net to adequately learn the distribution of the problem set. Is there a rule of thumb on estimating the size of required training data?

It would be nice to see some ablation test results that will display the increase in the performance of the L2P+MIP and eventual saturation as more and more training data is used in constructing the neural net model.

**Questions:**

Questions are described in the weakness section.

---

> ### Author Response · Authors · 2023-11-18
> **Response to Reviewer hVoV**
>
> We appreciate the reviewer for the comments on our paper. Thank you for your acknowledgment of our work. We set out below our responses to each of the questions you may concern about.
> >***Q1: Details of the neural net training procedures.***
> >
> > Sorry for the inconvenience, we now add more details of experimental settings in Section 4.1. For the train/test size. We use the train/test split rate as 80%/20% for all datasets. For the easy datasets, we generate 1000 instances for each. For the two industrial datasets, they contain 1000 instances for each. For the medium and hard datasets, we directly split the instances provided by their original benchmarks. Here we list these datasets and their total number of instances: Corlat(2000) [1], MIK(100) [2], Item Placement(10000) [3], Load Balancing(10000) [3], Anonymous(118) [3]. The usage of these datasets is common in existing papers and we just follow them [4,5,6,7]. The number of epochs for training is 10,000. The time of SA to provide a solution for each instance is hours in the medium datasets and days in the hard datasets. Since the training part is totally offline, thereby we do not focus on its efficiency. In the running step, for each instance, our L2P needs less than 0.05s in the medium datasets, and less than 0.5s in the hard datasets, which is efficient enough for practical usage.
> >
> > ***Q2: Is there a rule of thumb on estimating the size of required training data?***
> >
> > We appreciate your kind reminder. For the number of data for training, we directly follow the settings in existing work [4,5,6,7], without any specialized optimization. Admittedly, finding the most suitable number of training data to adequately learn is meaningful in the sense of avoiding overfitting. In our reply to your next question, we managed to conduct a series of experiments, trying to release your concerns. As far as we observed, the current need for estimating the size of required training data is not urgent in the field of ML-for-MIP, and we will continue to focus on this topic in our future work.
> >
> > ***Q3: Ablation test about adjusting the training data.***
> >
> > Thank you for your valuable suggestion. We have conducted an ablation study by adjusting the training data to 20%, 40%, 60%, and 80% of the original training data. The table below shows the results of the MIRP and CorLat datasets. We plot several figures to better show the trend of L2P and other baselines in Appendix(A.9, Figure 6, page 17) of our revised pdf, please refer to it.
> >  |   Ratio of training data   |  20% | 40%   | 60% | 80% | 100% |
> > | - | - | - | - | - | - |
> > | MIRP|  4.18%   | 7.74% | 10.11%|   10.87%    | 11.12%|
> > | CorLat    | 8.27%  | 23.17% |  30.37%     | 33.28% |34.74%|
> >
> > We can see that with more training data used, L2P will perform better. But we also notice that after using 80% of the data, the performance of L2P tends to stabilize. Therefore, the review's idea of finding the suitable data size for training indeed makes sense, and we will keep following.
>
>
> [1] Carla P Gomes, Willem-Jan van Hoeve, and Ashish Sabharwal. Connections in networks: A hybrid approach. In International Conference on Integration of Artificial Intelligence (AI) and Operations Research (OR) Techniques in Constraint Programming, pp. 303–307. Springer, 2008
>
> [2] Alper Atamt¨urk. On the facets of the mixed–integer knapsack polyhedron. Mathematical Programming, 98(1):145–175, 2003
>
> [3] Maxime Gasse, Simon Bowly, Quentin Cappart, Jonas Charfreitag, Laurent Charlin, Didier Chételat, Antonia Chmiela, Justin Dumouchelle, Ambros Gleixner, Aleksandr M Kazachkov, et al. The machine learning for combinatorial optimization competition (ml4co): Results and insights. In NeurIPS 2021 Competitions and Demonstrations Track, pp. 220–231. PMLR, 2022.
>
> [4] Exact Combinatorial Optimization with Graph Convolutional Neural Networks. Maxime Gasse, Didier Chételat, Nicola Ferroni, Laurent Charlin, Andrea Lodi NeurIPS 2019.
>
> [5] Han, Qingyu, et al. "A GNN-Guided Predict-and-Search Framework for Mixed-Integer Linear Programming." The Eleventh International Conference on Learning Representations. 2022.
>
> [6] Wang, Zhihai, et al. "Learning cut selection for mixed-integer linear programming via hierarchical sequence model." arXiv preprint arXiv:2302.00244 (2023).
>
> [7] Li, Sirui, et al. "Learning to Configure Separators in Branch-and-Cut." arXiv preprint arXiv:2311.05650 (2023).

---

> > ### Comment · Reviewer_hVoV · 2023-11-18
> >
> > Thank you for providing detailed responses to my concerns. I am impressed with the improved quality of the paper, particularly in the Experiment section.
> >
> > I have an additional concern to address. Could the paper include a discussion on the results for the 'Hard: Load Balancing' and 'Hard: Anonymous' problems? I apologize if this has already been discussed, but the observation that L2P outperforms SA seems counterintuitive. This could potentially indicate a flaw in the experimental methodology.

---

> ### Author Response · Authors · 2023-11-18
>
> Thank you for your prompt reply and the acknowledgment of our rebuttal work. We also notice the results for the 'Hard: Load Balancing' and 'Hard: Anonymous'. Though our method L2P is guided by SA, L2P did not mean to be always under SA since the testing set does not overlap with the training set, and the characteristic of the neural networks could make it learn better, e.g. the generalization. Therefore, the output presolving of L2P on the testing set may be different from the results of SA, not like in the training. As the searching space of presolving is very large, there are many local optimal points. As a result, SA and L2P may find different presolving points both better than the default presolving, which is very likely to happen due to the vast search space.
>
> For the Load Balancing dataset, actually the average improvement of L2P does not outperform SA, only in the effectiveness part. It is because in some instances the output presolving of L2P happens to improve the PD integral a little, thereby influencing the effectiveness. We use effectiveness (win rate) as a metric because it is common in existing papers, but this kind of disturbance may not be easy to avoid, in this sense, the average improvement is more stable to evaluate the performance.
>
> As for the 'Hard: Anonymous' dataset, this dataset only has 118 instances in total and we use 20 for evaluation, and the split of train and test is provided by the [benchmark](https://www.ecole.ai/2021/ml4co-competition/#datasets), not by ourselves. It seems that 11 instances of them have space to be better presolving, and all methods tend to improve them and end up with the same 55% effectiveness. We find that SA does not perform well on this dataset since both SMAC3 and our L2P outperform it in the average improvement. Due to the large scale of this dataset, the running time of SA is tremendous and may happen to miss the better solution. The few number of instances in this dataset may enhance this fluctuation, such as SA just working poorly on these 20 testing instances other than the training instances.
>
> As we mentioned before, the performance of SA on the testing set is not directly related to the performance of our L2P. Due to its tremendous time consumption, the performance of SA may vary and is not suitable for a baseline, as other methods (Random/SMAC3/FBAS/L2P) have a significantly small time consumption enough for practical usage. Admittedly, this phenomenon is interesting and we will try to explore it more in our future work.

---

> > ### Comment · Reviewer_hVoV · 2023-11-20
> >
> > I change my score as my concerns are addressed adequately.

---

> > > ### Author Response · Authors · 2023-11-20
> > >
> > > Thank you! We will continue to polish our paper in the final version, thanks again for your valuable comments.

---

### Official Review · Reviewer_32Tw · 2023-10-30

**Soundness:** 3 good
**Presentation:** 4 excellent
**Contribution:** 3 good
**Rating:** 6
**Confidence:** 4

**Summary:**

The authors propose a learning-based approach for selecting the priority, max rounds, and timing of presolvers for MIP solving. They do this by obtaining high quality presolver settings using simulated annealing for several training instances, then training a neural network to predict the high-quality setting for unseen test instances. The network architecture first predicts the priority then based on the priority prediction makes predictions for the max rounds and timing. They evaluate on a variety of settings that demonstrate their method gives improved performance over other hyperparameter prediction methods especially on hard MIP instances. The authors further perform an ablation study to determine the effectiveness of different components of their model and sensitivity to data collection method. Lastly, they investigate their resulting predictions to identify the impact of setting the individual hyperparameters. The paper is overall well written and they tackle an interesting problem of using machine learning to assist in presolving MIP instances.

**Strengths:**

The main strength of this work is that they tackle a new problem in learning-accelerated optimization, namely learning to improve presolving of MIP instances. Additionally, their results seem to give improved performance over baseline hyperparameter tuning methods and improve runtime of optimization solvers overall. Furthermore, the authors conduct extensive evaluation of their approach to better understand where the performance improvement comes from and how sensitive performance is to modification of their approach. Lastly, the authors evaluate their approach on a variety of domains that are used in real-world settings including sustainable corridor optimization, maritime inventory routing, load balancing, and item placement which have practical impacts, and which the authors demonstrate improved performance.

Additionally, the authors release their code both as a way to benchmark against their approach, but also as a way to enable others working in the space of predicting performant presolver parameters.

**Weaknesses:**

One weakness is that their approach doesn’t go beyond tuning the hyperparameters of the presolving methods. While their approach does give good performance and is a good first step towards using learning for presolving, it would be interesting to see methods that dive deeper, such as using machine learning to determine new effective methods for presolving.

**Questions:**

Simulated annealing also generates several examples of parameters and their values during the search process; however, that seems to be currently thrown away. Is there a way to use something like contrastive learning to learn from this thrown away data?
Similarly, is there some way to benefit from the fact that the “shape” of the parameter settings is the same throughout the experiment and don’t change from one instance to the next? Is it possible to re-evaluate several performant hyperparameter settings from one instance on a new instance to quickly collect data?

For hard/anonymous instances why is the 55% bolded when the other methods also have 55%?

---

> ### Author Response · Authors · 2023-11-18
> **Response to Reviewer 32Tw**
>
> Thank you for your valuable comments. We try to answer the questions and hope we can alleviate your concerns.
> > ***Q1: It would be interesting to see methods that dive deeper.***
> >
> > We appreciate your suggestion and we also consider that we should take a deeper look at presolving. Regrettably, we can not determine new effective methods for presolving. As far as we know, there is no prior work for using machine learning to discover new presolvers which we believe is valuable yet remains an open problem. *Nevertheless, we can try to utilize our L2P to find the effective parts inside existing presolving to some extent.*
> >
> > In Figure 9 on page 20 of our submitted paper, we show the illustrations for improved presolving parameters of different presolvers under multiple datasets. We can see parameters of the improved presolving vary a lot, especially the priority (the order of the presolvers). Meanwhile, we can find that different problem classes show different patterns: some approximately wave-like patterns in each figure. For example, presolver 7 10 11 seem to take more responsibility, as the priority of them tends to be higher. However, in other datasets they become less important, especially in the Anonymous dataset presolver 7 is almost the last presolver to be deployed. In this sense, our L2P can manage to find the effectiveness part in presolving. Admittedly, we consider your suggestion about finding more effective presolving methods to be really important to this area, and we will keep working on it.
> >
> > ***Q2: Use something like contrastive learning.***
> >
> > Thank you for your valuable suggestion. As you suggested, the non-optimal results of SA can be regarded as negative samples and the optimal results are positive samples. We did not think of this idea before since we are not familiar with contrastive learning. After our full survey of this topic, especially work from InfoNCE[1], we find one existing work[2] that shares a similar idea, where a contrastive loss is used to encourage the model the distinguish the positive/negative samples, and make predictions similar to the positive samples. We modify their contrastive loss and average it with our original loss after dynamic loss averaging. We test the performance with the variance of the ratio of contrastive loss ranging from 0.0 to 0.2, where 0.0 denotes the original L2P.
> > |   Ratio of contrastive loss   |  0.0 | 0.05   | 0.1 | 0.15 |0.2 |
> > | - | - | - | - | - | - |
> > | CorLat |  34.74%   | 34.97% |  32.18%|   29.87%    |  25.59% |
> > | MIK |   3.02%  | 2.99% |  2.77% |   2.52%   | 2.11% |
> >
> > We can see that the improvement of adding contrastive loss is not as significant as we expected. It may be because our dynamic loss averaging module is not designed with the consideration of an extra contrastive loss. Moreover, the outputs of SA vary a lot during the searching procedure, which may disturb the distribution of negative samples. Above all, the idea of using contrastive learning does make sense, but it may be incompatible with the current design for L2P, which needs further research and experiments to find out.
> >
> > ***Q3: Re-evaluate based on the similarity of parameters on different instances to quickly collect data.***
> >
> > Thank you for your good question. As we mentioned before, the parameters of the improved presolving vary a lot even if the instances are from the same category, as shown in Figure 9 on page 20 of our submitted paper. In the MIP instances, a tiny variance of the coefficients may result in different performance. Therefore, we consider the implementation of re-evaluate is non-trivial. Actually, as the collection of data by SA is offline, the need to improve its efficiency is not urgent to us. We will keep tracking it if a scenario need such property occurs in the future.
> >
> > ***Q4: Mistake in the table.***
> >
> > Sorry for the confusion. We have refined the pdf, please refer to it.
>
> [1] Oord, Aaron van den, Yazhe Li, and Oriol Vinyals. "Representation learning with contrastive predictive coding." arXiv preprint arXiv:1807.03748 (2018).
>
> [2] Huang, Taoan, et al. "Searching large neighborhoods for integer linear programs with contrastive learning." International Conference on Machine Learning. PMLR, 2023.

---

### Official Review · Reviewer_zK3o · 2023-10-31

**Soundness:** 3 good
**Presentation:** 3 good
**Contribution:** 4 excellent
**Rating:** 6
**Confidence:** 3

**Summary:**

This paper proposes Learning to Presolve (L2P) for Mixed Integer Programming (MIP). The presolving of the MIP solver is a routine that simplifies the MIP instance, enabling the succeeding main solving routine (e.g., B&B) to operate more efficiently. Current presolving routines are hard-coded and rely on expert knowledge, making them unable to account for instance-wise differences in MIP problems. L2P employs a supervised learning method to predict the best presolver routine while using the result of a powerful yet time-consuming search method as the training label.

Using the trained model, L2P proposes tailored presolving routines for each instance in fractions of a second. Such tailored routines lead to a significant improvement in MIP solving performance. The proposed L2P consistently demonstrates performance gains in various MIP datasets while requiring a negligible amount of presolving parameter suggestions.

**Strengths:**

- The writing is easy to follow and effectively conveys the necessary information to understand the manuscript.
- L2P addresses an understudied aspect of MIP by proposing appropriate presolving parameters on an instance-wise basis.
- L2P consistently demonstrates performance improvements in empirical experiments.

**Weaknesses:**

- While the current L2P model shows satisfactory improvements over the baselines, further performance gains may be achievable by leveraging more modern neural network architectures. For example, the authors used GCNN as the backbone of the feature-extracting GNN, which could potentially be replaced with graph-formers in a drop-in replacement manner.
- It may be beneficial to include some related work in the literature review and experiment section. For example, I had the opportunity to review a paper titled "Accelerate Presolve in Large-Scale Linear Programming via Reinforcement Learning," which addresses the same problem but in the LP domain.

**Questions:**

- Has the performance gain observed from SCIP also been found with other MIP solvers such as GUROBI and CPLEX?
- Regarding the training of the L2P model, how crucial are KRL and dynamic loss averaging? Do they significantly affect the overall trend of the results?

---

> ### Author Response · Authors · 2023-11-18
> **Response to Reviewer zK3o (1/2)**
>
> Thank you for the time, thorough comments, and nice suggestions. We answer the comments and questions point-by-point, clarify the ablation experiments and discuss with the related work.
>
> > ***Q1: Leveraging more modern neural network architectures.***
> >
> > Thanks for this valuable suggestion. We did not think of this point before since GCNN is commonly used in the related work while proposing better GNN architecture is not the focus of this paper. In fact, this is also the case for the ML-for-MIP community whereby the standard GNN is largely used in literature, and there even exists papers challenging the role of GNN in solving combinatorial problems [7]. Hence how to effectively leverage GNNs for MIP is still an open problem.
> >
> >On the other hand, we notice that RandomGNN [1] is a tailored approach, to add different special features to each node in the process of GNN. It can make GNN better distinguish the nodes and enhance the representation of the MIP instance. We consider this approach could be a possible improvement to existing GNN for our problem. Moreover, as the reviewer mentioned, we try to test the performance of GATConv [2] and TransformerConv [3] as drop-in replacements.
> > |      | Original L2P | L2P + RandomGNN[1] | L2P + GATConv[2]   | L2P + TransformerConv[3]|
> > | - | - | - | - | - |
> > | CorLat | 34.74%    | 35.11% |  34.92%|   35.36%    |
> > | MIK |   3.02%  | 3.08% | 2.87% |   3.03%   |
> >
> > Due to the short time window for response in ICLR 2024, we first test the performances of different GNN backbones on the medium datasets CorLat and MIK. It turns out the replacements of the GNN backbone can actually improve the performance of L2P, but only to a small extent. It may be because the processing of features after GNN is more valuable in L2P. We appreciate the review for the idea of adding a drop-in replacement for the GNN backbone, and we will continue to work in this direction.
> >
> > ***Q2: About one concurrent work.***
> >
> > Thanks for your reminder, we are sorry that we did not include this paper in our submitted paper, since it was recently submitted to Arxiv after our ICLR submission. We find that this concurrent work also introduces a new method utilizing reinforcement learning for presolve, while they focus on the LP domain unlike us on the MIP domain. In fact, solving MIP is different from solving LP in many aspects [8]. For example, the branch-and-bound algorithm in MIP involves solving large batches of similar problems efficiently, and the cutting-plane method in MIP adds linear inequalities to the original problem to eliminate fractional solutions. These techniques are easily influenced by the presolving of MIP, but not required by LP.
> >
> > Specifically, they use RL to tackle the presolve decisions in the process of solving LP. Since the solving of MIP in our work is more complicated than LP, their method may focus on different aspects of the solvers, like the termination point which is critical in LP solvers. However, due to the inefficient sample utilization in RL, the proposed method in this concurrent work may need more time in trial-and-errors and still faces the challenges of sparse reward during training.
> >
> > Compared to this work, our L2P can take full advantage of samples collected in offline training and the online testing process is more stable and efficient, which is important in practical usage in our opinion. We appreciate the reviewer for mentioning this work, and we have included it in the related work of our revised paper and are thinking of further interactions with our work in the future.
> >
> > ***Q3: Performance about other MIP solvers such as GUROBI and CPLEX.***
> >
> > Thanks for your valuable question. Many commercial MIP solvers such as Gurobi and CPLEX are also equipped with kinds of presolving techniques. However, their APIs to adjust the presolving are not exposed since they are commercial solvers and not open-source, and that's why we can not test our proposed method on the commercial solvers. We consider it impossible to directly improve commercial solvers in research papers, as existing papers [4,5,6] mainly conduct experiments on open-source solvers (SCIP) like us. If we are able to cooperate with the commercial solver company in the future, we can also add our method to their internal strategy, integrating our machine learning technologies to help users better solve MIPs.

---

> ### Author Response · Authors · 2023-11-18
> **Response to Reviewer zK3o (2/2)**
>
> >
> > ***Q4: How crucial are KRL and dynamic loss averaging?***
> >
> > Sorry for the confusion, we have conducted an ablation study in Section 4.4 and Figure 4 of the submitted paper. As shown in Figure 4, we remove the key components of our L2P one by one, which begins with dynamic loss averaging and KRL. As shown in the figure, removing the dynamic loss averaging and KRL will result in a nearly 5% and 10% improvement drop in CorLat, while the total improvement is merely about 35%. We are sorry that we did not emphasize enough about this ablation study. We have refined this section in our paper and please refer to it.
>
>
> [1] Chen, Ziang, et al. "On Representing Linear Programs by Graph Neural Networks." ICLR 2023.
>
> [2] Brody, Shaked, Uri Alon, and Eran Yahav. "How attentive are graph attention networks?." ICLR 2021.
>
> [3] Shi, Yunsheng, et al. "Masked label prediction: Unified message passing model for semi-supervised classification." IJCAI 2021.
>
> [4] Exact Combinatorial Optimization with Graph Convolutional Neural Networks Maxime Gasse, Didier Chételat, Nicola Ferroni, Laurent Charlin, Andrea Lodi NeurIPS 2019.
>
> [5] Pochet Y, Wolsey L A. Production planning by mixed integer programming. New York: Springer, 2006.
>
> [6] Laporte G. Fifty years of vehicle routing. Transportation science, 2009, 43(4): 408-416.
>
> [7] What's Wrong with Deep Learning in Tree Search for Combinatorial Optimization, ICLR 2022
>
> [8] https://www.mathworks.com/help/optim/linear-programming-and-mixed-integer-linear-programming.html

---

> > ### Comment · Reviewer_zK3o · 2023-11-21
> > **Reviewer Response**
> >
> > Thank you for providing a response to my review! I greatly appreciate the updates as they have been helpful in enhancing my understanding of the paper. As I mentioned earlier in my review, I believe the paper meets the standards expected at ICLR. However, it may still require further improvements to be considered a strong paper. In that regard, I'd like to keep my initial evaluation, and I have no objection to the acceptance of this paper.

---

> > > ### Author Response · Authors · 2023-11-22
> > >
> > > Thank you for your acknowledgment of our work. We really appreciate your support.

---

### Official Review · Reviewer_iKXb · 2023-11-04

**Soundness:** 3 good
**Presentation:** 2 fair
**Contribution:** 3 good
**Rating:** 6
**Confidence:** 4

**Summary:**

The paper presents an innovative framework for Learning to Presolve (L2P) within the context of Mixed Integer Programming (MIP) solving using deep learning. To my knowledge, deep learning methods to solve MIPs haven’t explored L2P. Therefore, this work introduces a novel area of research. The proposed framework is well-conceived, offering a new perspective on presolving, and the evaluation suggests that deep learning can tailor presolving strategies to individual MIP instances, potentially enhancing solver efficiency.

**Strengths:**

1. **Novel Area of Research:** The paper investigates a novel and promising application of deep learning to improve MIP solving, a problem that has not been significantly explored before.

2. **Framework Development:** The authors have developed a new framework to systematically address the identified research gap. This framework could serve as a foundational benchmark for future research in this area.

3. **Empirical Evaluation:** The proposed framework has been rigorously evaluated, with results demonstrating its potential utility for the MIP solving community, which could lead to more targeted and efficient presolving methods.

**Weaknesses:**

1. **Lack of Detail in Methodology:** The manuscript does not adequately detail critical components of the proposed methods, such as the loss function and the domain of labels—information which is relegated to the Appendix. Moreover, key metrics like the PD integral lack a clear mathematical definition.

2. **Reporting of Metrics:** The paper uses the arithmetic mean to report solving times, which could be skewed by outliers. The geometric mean is a standard in the MIP community for its robustness to extreme values, and its absence is felt in the current reporting.

3. **Comparison and Ablation Studies:** The presolving time is only provided for the L2P method. For a comprehensive understanding of efficiency, it would be beneficial to see these times for other compared methods. Additionally, the specific benefits of KRL are not clear. An ablation study to evaluate the impact of KRL would be informative.

4. **Experimental Details:** The number of instances evaluated in Table 1 is unclear. More detailed comparisons, including the time taken during the Running Step for each method, would strengthen the results section. In Table 4, the metric used and the number of instances in each problem family should be stated for clarity.

While the authors have laid a robust foundation with their proposed framework, addressing a significant challenge within the field, I believe further refinement is necessary prior to its wider introduction to the research community. Enhancements such as additional methodological details, comprehensive ablation studies, and the adoption of standard metrics are essential to fully realize and validate the framework's potential.

**Questions:**

1. **Default Parameters:** For reproducibility, could the default parameters (42 x 3) be listed explicitly in the Appendix (Table 5)?

2. **Neural Network Representation:** Figure 3 is unclear as it suggests that a neural network is derived from an MLP, which is already a type of neural network. Could this be clarified?

3. **Graph Embedding Features:** How are these computed and what aggregation mechanism is used? Is there an aggregation of embeddings across all variables?

4. **Priority Label Input:** Have the authors considered using Priority Label as an input, akin to teacher forcing, and could this approach be compared with the current methodology?

5. **Dynamic Loss Averaging:** More details on this, including how weights are adjusted based on convergence, would be beneficial. If a library function is used, please reference it.

6. **Section Clarity:** In Section 4.2, the fifth sentence needs grammatical correction. Additionally, could Section 4.3 further elaborate on the potential privacy issues mentioned?

7. **PD-Integral:** A mathematical description of the PD-integral and its expected range would greatly aid in understanding its application and interpretation.

8. **Table Details:** In Table 4, the metric used and the number of instances in each problem family should be stated for clarity.

**Typos and Corrections:**

1. **Appendix Repetition:** There is a paragraph repeated in Appendix A.1 that should be corrected.

2. **SA Method clarification:** In Appendix A.2, following Equation 4, it seems that $\Delta_y$ should correspond to the time to solve the instances, as this is the primary optimization concern. Please confirm if this is the case.

---

> ### Author Response · Authors · 2023-11-18
> **Response to Reviewer iKXb (1/3)**
>
> Thank you for the time, thorough comments, and nice suggestions. We answer the comments/questions point-by-point, clarify our experiment setting and refine our paper as suggested.
> >
> >***Q1: Lack of details***
> >
> >We are sorry for the confusion. The loss functions used in our methods are ListMLE for the priority and Cross-Entropy for the max-round and timing. ListMLE [1] is a loss designed for ranking, which is suitable for learning the priority since the priority denotes the order/rank of the presolvers. We understand that placing the details of the experiment in the appendix is not recommended, therefore, we have reorganized the paper structure and enriched Section 4.1 to make our experimental settings more clear, sorry again for the inconvenience.
> >
> >For the definition of PD integral, the primal-dual integral in the field of mixed integer programming (MIP) is a measure used to evaluate the performance of MIP solvers. It provides a quantitative assessment of how effectively a solver bridges the gap between the primal (the best feasible solution found) and dual (a bound on the optimal solution) bounds over the course of the solution process. For its mathematics formulations, we recommend the readers to read the [documentation](https://www.ecole.ai/2021/ml4co-competition/\#metrics) of one benchmark we used. The metric section of this documentation describes the calculation of PD integral with informative figures. Here we list a more detailed breakdown, and we also place it in our revised paper:
> >1) Primal and Dual Solutions in MIP: In mixed integer programming, a primal solution refers to a feasible solution that satisfies all the constraints of the MIP model. The dual solution, on the other hand, is related to the bounds on the optimal solution value. In linear programming, this would be equivalent to the solution of the dual problem, but in MIP, it usually refers to a bound derived from a relaxation of the integer constraints.
> >2) Gap Measurement: The primal-dual integral measures how the gap between the primal and dual solutions evolves over time. This gap is the difference between the objective value of the best known feasible solution (primal) and the best known bound (dual).
> >3) Purpose of the Primal-Dual Integral: This measure is especially useful in understanding the efficiency of a solver in converging to the optimal solution. It can highlight whether a solver quickly identifies good feasible solutions and tight bounds or whether it struggles to improve the primal and dual solutions over time.
> >4) Calculation: The primal-dual integral is calculated by integrating the gap over the time taken for the computation. A lower integral value indicates a more effective solver, as it shows that the solver was able to keep the primal-dual gap smaller throughout its run.
> >5) Use in Analysis and Benchmarking: This metric is valuable for comparing different solvers or solution strategies in mixed integer programming. It provides insights beyond just the final solution quality or computation time, focusing on the overall trajectory of the solution process.
> >
> >***Q2: Reporting of metrics.***
> >
> >Thanks for your kind reminder. Here, we consider the geometric mean may not be suitable for the task in our paper. In our learning to presolve task, we focus on the solving time/PD integral improvement compared to the default via presolving instead the solving time/PD integral itself. Therefore, we care for the average improvement, while we do not care whether the actual value of the solving time/PD integral is small or large, which already avoids the case of outliers if an instance is too simple or too hard. Moreover, the solving time/PD integral improvement of each instance could be zero or negative, while the geometric mean requires all terms to be positive.
> >
> >***Q3: Comparison and ablation studies.***
> >
> >Thanks for your suggestion. For the running time of methods, L2P/FBAS/SMAC3 shares a similar running time. For each instance, they need less than 0.05s in the medium datasets, and less than 0.5s in the hard datasets, while the SA needs hours in the medium datasets and days in the hard datasets. Here what we want to claim is that the consumption of SA is unacceptable. Since the time usages of L2P/FBAS/SMAC3 are already enough for practical usage, we consider it acceptable that our L2P does not have a significant speed advantage over baselines.
> >
> >As for the ablation study of KRL, actually we have an ablation study in Section 4.4 and Figure 4 of the submitted paper. As shown in Figure 4, we remove the key components of our L2P one by one, which begins with dynamic loss averaging and KRL. As shown in the figure, removing the dynamic loss averaging and KRL will result in a nearly 5% and 10% improvement drop in CorLat, while the total improvement is merely about 35%. We are sorry that we did not emphasize enough about this ablation study. We have refined this section in our uploaded paper.
> >

---

> > ### Comment · Reviewer_iKXb · 2023-11-20
> > **Some more clarifications**
> >
> > Thank you very much for the clarifications.
> >
> > Regarding the three tasks you examined—priority, max rounds, and timing—I am interested in knowing more about the domain values for these variables. Specifically, what are the range and nature of the values used for labels in these tasks? Could you also elaborate on the type of loss function utilized for each of these tasks? I apologize if this detail was mentioned and I overlooked it.
> >
> > **Suggestion:**
> > I would also like to suggest a minor improvement for future iterations of your paper. In Figures 4 and 6, the annotations were quite challenging to read due to the small font size. Using a bigger font for these figures could significantly enhance their clarity and impact.

---

> ### Author Response · Authors · 2023-11-18
> **Response to Reviewer iKXb (2/3)**
>
> >
> >***Q4: Experimental details.***
> >
> > Sorry for the confusion, we have refined Section 4.1 and clarified our experimental settings. Specifically: For the easy datasets, we generate 1000 instances for each. For the two industrial datasets, they contain 1000 instances for each. For the medium and hard datasets, we directly split the instances provided by their original benchmarks. Here we list these datasets and their total number of instances: Corlat(2000) [2], MIK(100) [3], Item Placement(10000) [4], Load Balancing(10000) [4], Anonymous(118) [4]. The usage of datasets is common in existing papers and we just follow them [5,6,7,8]. The running time is shown in Q3. As for Table 4, the settings are the same as the main experiments in Table 1, including the number of instances and the metric. It was our negligence and we have improved the description of datasets (Section 4.1.1) in our revised paper.
> >
> >***Q5: Default parameters.***
> >
> > Thank you for your advice, we have placed the default parameters in Table 9, please refer to it.
> >
> >***Q6: Neural network representation.***
> >
> > Sorry for the confusion. Here the arrow in Figure 3 denotes the flow of data and how it passes through the inside modules in our framework, we did not mean to derive a network. We have refined it in its caption.
> >
> > ***Q7: Graph embedding features.***
> >
> > For the GNN backbone, we just follow the GCNN[5], where the embedding is aggregated across all variables. This GNN backbone is commonly used in the literature[6][7][8] so we did not detailed introduce it in our paper, sorry for the inconvenience. Inspired by other reviewers, we add experiments for trying different GNN backbones, please refer to Q1 of the response to Reviewer zK3o.
> >
> > ***Q8: Priority label input.***
> >
> > Thanks for your suggestion. During our attempts, we tested using ground truth priority as labels to see the training results of max-rounds and timing. As a result, the loss of learning max-rounds and timing goes down quicker than before. However, this approach can not be used in the real world since it needs extra ground truth priority information, which leads to an unfair comparison to the methods in our paper. Therefore, we do not regard this method as a baseline.
> >
> > ***Q9: Dynamic loss averaging***
> >
> > Sorry for the confusion. We now add more details in Appendix A.5. Specifically, imbalance among multiple tasks is one of the most critical issues to be addressed in multi-task learning, many studies have attempted to balance the convergence rate of different tasks by assigning an adaptive weight to each task. [9,10,11]
> Inspired by them, the motivation of our Dynamic Loss Average method is to average task weighting over time via adjusting the rate of change of loss for each task. Formally, we define a loss weight: $\omega_k$ for each task: $k$ as:
> $$\omega\_k(t):=\frac{K exp(\mathcal{R}\_k(t-1)/\mathcal{T})}{\sum_{i}{exp(\mathcal{R}_i(t-1)/\mathcal{T})}},$$
> $$\mathcal{R}\_k(t-1)=\frac{\mathcal{L}\_{k}(t-1)}{\mathcal{L}\_{k}(t-2)}$$
> where $\mathcal{L}_k(t)$ denotes the loss of task $k$ at the $t$ iteration and $\mathcal{R}_k(\cdot)$ calculates the relative training convergence rate of task $k$, $\mathcal{T}$ in the softmax operator represents the temperature that controls the softness of weighting for task-specific loss. In the implementation, temperature $\mathcal{T}$ is set to 2 and $\mathcal{R}_k(t)$ is initialized as 1 when $t=1,2$, but any other effective initialization methods with prior knowledge can also be applied here.
> >
> > ***Q10: Other mistakes in the paper.***
> >
> > Thanks for your kind reminder. We have further refined our paper, added the description of PD integral (mentioned by Q1), described the setting in Table 4, corrected several sentences, reorganize the sturture of Section 4 (Experiments), and improved the text of the appendix. For the potential privacy issue mentioned in Section 4.3, it is due to the policy of the company that provides us with private industrial datasets. Due to its policy, its dataset can only be used on their computer, and the code for running must also be authorized. Therefore, deploying SMAC3 and FBAS on their computers may be troublesome. In this sense, the potential privacy is more like business sensitivity.

---

> ### Author Response · Authors · 2023-11-18
> **Response to Reviewer iKXb (3/3)**
>
> [1] Xia, Fen, et al. "Listwise approach to learning to rank: theory and algorithm." Proceedings of the 25th international conference on Machine learning. 2008.
>
> [2] Carla P Gomes, Willem-Jan van Hoeve, and Ashish Sabharwal. Connections in networks: A hybrid approach. In International Conference on Integration of Artificial Intelligence (AI) and Operations Research (OR) Techniques in Constraint Programming, pp. 303–307. Springer, 2008
>
> [3] Alper Atamt¨urk. On the facets of the mixed–integer knapsack polyhedron. Mathematical Programming, 98(1):145–175, 2003
>
> [4] Maxime Gasse, Simon Bowly, Quentin Cappart, Jonas Charfreitag, Laurent Charlin, Didier Chételat, Antonia Chmiela, Justin Dumouchelle, Ambros Gleixner, Aleksandr M Kazachkov, et al. The machine learning for combinatorial optimization competition (ml4co): Results and insights. In NeurIPS 2021 Competitions and Demonstrations Track, pp. 220–231. PMLR, 2022.
>
> [5] Exact Combinatorial Optimization with Graph Convolutional Neural Networks. Maxime Gasse, Didier Chételat, Nicola Ferroni, Laurent Charlin, Andrea Lodi NeurIPS 2019.
>
> [6] Han, Qingyu, et al. "A GNN-Guided Predict-and-Search Framework for Mixed-Integer Linear Programming." The Eleventh International Conference on Learning Representations. 2022.
>
> [7] Wang, Zhihai, et al. "Learning cut selection for mixed-integer linear programming via hierarchical sequence model." arXiv preprint arXiv:2302.00244 (2023).
>
> [8] Li, Sirui, et al. "Learning to Configure Separators in Branch-and-Cut." arXiv preprint arXiv:2311.05650 (2023).
>
> [9] Kendall, Alex, Yarin Gal, and Roberto Cipolla. "Multi-task learning using uncertainty to weigh losses for scene geometry and semantics." Proceedings of the IEEE conference on computer vision and pattern recognition. 2018.
>
> [10] Chen, Zhao, et al. "Gradnorm: Gradient normalization for adaptive loss balancing in deep multitask networks." International conference on machine learning. PMLR, 2018.
>
> [11] Liu, Shikun, Edward Johns, and Andrew J. Davison. "End-to-end multi-task learning with attention." Proceedings of the IEEE/CVF conference on computer vision and pattern recognition. 2019.

---

> ### Author Response · Authors · 2023-11-20
>
> Thank you for your prompt reply and kind suggestions. Here we will describe the domain values and loss of the priority, max-round, and timing in our task, based on the official documentation of [SCIP](https://www.scipopt.org/doc/html/PARAMETERS.php) (you can search for the word "presolver" on the linked website to find related information).
> - Priority: The range of priority of SCIP is [-536870912,536870911], as what we care about is the order instead of the value itself, we just let the neural network output real numbers and use the ListMLE loss to guide the order of them. ListMLE [1] is a loss designed for ranking, which is suitable for learning the priority since the priority denotes the order/rank of the presolvers. It uses a permutation probability model to map a list of scores to a probability distribution and then minimizes the negative log-likelihood loss between the predicted distribution and the ground truth distribution. We recommend the official [documentation](https://www.tensorflow.org/ranking/api_docs/python/tfr/keras/losses/ListMLELoss) to see its mathematical formulation, implementation, and examples.
> - Max-round: The range of max-round in SCIP is [-1,2147483647], while -1 denotes unlimited. For convenience, we set the max-round task as a multiclass classification in [-1, 0, 1, 2]. Though the range of max-round in SCIP is large, most default parameters are not larger than 1, since the larger values could be included by setting "-1" to some extent. As we formulate it as a multiclass classification, we use the [Cross-Entropy](https://ml-cheatsheet.readthedocs.io/en/latest/loss_functions.html) loss, which is common in multiclass classification.
> - Timing: The range of timing in SCIP is [4:FAST, 8:MEDIUM, 16:EXHAUSTIVE, 32:FINAL], which is already a multiclass classification, and we also use the Cross-Entropy loss.
>
> Though the number of classes in max-round/timing is not large, there are 14 presolvers and the total space is at least (14!) * 4^14 * 4^14 $\approx$ 6.28e+27. Here we use 14! to represent the space of priority but it is actually much larger than this since the output priority of the neural network is continuous value and not constrained.
>
> In our paper, we mentioned the loss functions in the experimental settings. For the domain of priority, max-round, and timing, we did not directly mention them, but you can find them in Figure 7 8 9 and Table 9, where we output the presolving parameters.
>
> For the annotations of Figures 4 and 6, we have modified Figure 6 in the new version of our paper, but for Figure 4 we may need more time to adjust the figure due to the space limit of the main paper. We will manage to fix this as soon as possible. Sorry for the inconvenience.
>
> [1] Xia, Fen, et al. "Listwise approach to learning to rank: theory and algorithm." Proceedings of the 25th international conference on Machine learning. 2008.

---

> > ### Comment · Reviewer_iKXb · 2023-11-21
> > **Thank you!**
> >
> > Thank you for your comprehensive and engaging responses to my queries. Your clarifications have significantly enhanced my understanding of your work. I have revised my scores upward to reflect the potential of your research.

---

> > > ### Author Response · Authors · 2023-11-21
> > >
> > > Thank you for your kind support!
> > > We have improved Figure 4 in the main paper as you suggested, please refer to it.

---

### Author Response · Authors · 2023-11-18
**Global Response to Chairs and Reviewers**

We thank all reviewers for their kind reviews and helpful comments. We are delighted with their encouraging comments:
1. The idea of learning to presolve is novel and important to this area. (iKXb, 32Tw, hVoV)
2. The proposed framework is well-designed and open-source, which could serve as a benchmark for further research. (iKXb, zK3o, 32Tw)
3. The experiments conducted on various datasets demonstrate the performance of the proposed method. (iKXb, zK3o, 32Tw, hVov)
4. The paper is easy to follow and understand. (zK3o, 32Tw)

During our rebuttal, we post detailed responses to each reviewer and try to answer their questions, ease their concerns, and further improve our paper. We thank the reviewers again for the valuable comments that are helpful in refining our paper. In particular, we reorganize Section 4 (Experiments) to make our experimental settings and results more clear. Please refer to our rebuttal under each official review and the revised PDF.

---

### Meta-Review · Area_Chair_zHTZ · 2023-12-06

**Metareview:**

(a) Summarize the scientific claims and findings of the paper based on your own reading and characterizations from the reviewers.
- The authors explore the problem of presolving of MIP instances.
- They suggest replacing standard (generic) MIP presolving by (tailored) ML method
- This type of idea has shown potential for MIP solving but is, for the first time, used at resolving time
- The authors design an ML approach that learns to replicate the simulated annealing heuristic (used as a hyper-parameter opt. method)
- The results show that the proposed approach achieves a reasonable accuracy/time tradeoff compared to simulated annealing and are competitive (at least) compared to default heuristics used in an open-source solver

(b) What are the strengths of the paper?
- Manuscript is well written, easy to follow (I'd say even for someone not intimately familiar with MIPs/B&B)
- The method is sound and well-adapted to the problem
- Empirical comparisons across a set of synthetic and real-world instances

(c) What are the weaknesses of the paper? What might be missing in the submission?
-

**Justification For Why Not Higher Score:**

The reviewers were unanimous that this paper should be accepted. (Two reviewers even raised their scores after the author's response.)

The domain explored in this paper (ML for MIP presolving) is novel. Replacing a strong but slow heuristic with a fast neural network is becoming somewhat standard in ML for MIP solving (e.g., authors reuse the GNN encoding, the data generation methodology, and some of the experimental setup from the literature). As such, while the paper's results are exciting and might lead to others exploring this problem (presolving), this work is likely not ground-breaking for machine learning.

ML for combinatorial optimization is an important field (e.g., improving CO likely has clear and measurable industrial balue), but perhaps it is also a bit niche in a venue such as ICLR.

As such, I find it reasonable to accept the paper as a poster.

**Justification For Why Not Lower Score:**

This work is sound and the paper is likely to be interesting to the ICLR community. All reviewers agree that it's above the acceptance threshold.

---

### Decision · Program_Chairs · 2024-01-16

Accept (poster)